# Evaluation of serum MMP-2 and MMP-3, synovial fluid IL-8, MCP-1, and KC concentrations as biomarkers of stifle osteoarthritis associated with naturally occurring cranial cruciate ligament rupture in dogs

**Sarah Malek[1]\*, Hsin-Yi Weng[2], Shannon A. Martinson[3], Mark C. Rochat[1], Romain Béraud[4], Christopher B. Riley[5]**

**1** Department of Veterinary Clinical Sciences, Purdue University School of Veterinary Medicine, West Lafayette, Indiana, United States of America, **2** Department of Comparative Pathobiology, Purdue University School of Veterinary Medicine, West Lafayette, Indiana, United States of America, **3** Department of Pathology and Microbiology, Atlantic Veterinary College, University of Prince Edward Island, Charlottetown, Prince Edward Island, Canada, **4** Centre Vétérinaire Daubigny, Quebec City, Quebec, Canada, **5** School of Veterinary Science, Massey University, Palmerston North, New Zealand

\* maleks@purdue.edu

## Abstract

The purpose of this study was to evaluate matrix metalloproteinases (MMP) -2 and MMP-3 in serum, and keratinocyte-derived chemoattractant (KC), interleukin 8 (IL-8) and monocyte chemoattractant 1 (MCP-1) in synovial fluid (SF) as stifle osteoarthritis (OA) biomarkers in dogs. Dogs with naturally occurring cranial cruciate ligament (CrCL) rupture (OA group) and healthy controls were recruited. Stifles with CrCL deficiency were surgically stabilized. Serum, SF, and synovial biopsy samples were collected from the OA group preoperatively, whereas samples were collected once from control dogs. A blinded veterinary pathologist graded synovial biopsies. Serum and SF analyses were performed using xMAP technology. General linear regression was used for statistical comparisons of serum biomarkers, and mixed linear regression for SF biomarkers and temporal concentration changes. The overall discriminative ability was quantified using area under curve (AUC). Spearman's correlation coefficient was used to assess correlations between synovial histology grades and the biomarkers. Samples from 62 dogs in the OA group and 50 controls were included. The MMP-2 and MMP-3 concentrations between the OA and control groups were not significantly different, and both with an AUC indicating a poor discriminative ability. All three SF biomarker concentrations were significantly different between the OA group and controls ($P < 0.05$). The MCP-1 was the only biomarker showing an acceptable discriminative performance with an AUC of 0.91 (95% confidence interval: 0.83–0.98). The sum of the inflammatory infiltrate score was significantly correlated with all three SF biomarkers ($P < 0.01$). Summed synovial stroma, and all scores combined were significantly correlated with IL-8 and MCP-1 concentrations ($P < 0.003$), and the summed synoviocyte scores were significantly correlated with

**Data Availability Statement:** All relevant data are within the paper and its Supporting Information files.

**Funding:** The funding for this study was secured through Canadian Institutes of Health Research Grant-Regional partnership fund - Innovation PEI (No: 97027) (CBR) (https://cihr-irsc.gc.ca/e/193.html); Companion Animal Trust Fund – University of Prince Edward Island (SM, RB) (https://www.upei.ca/avc/companion-animals/companion-animal-trust-fund); and the Cohn Family Chair for Small Animals- Oklahoma State University (SM) (https://news.okstate.edu/magazines/state-magazine/articles/2018/spring/cohn-family-chair-for-small-animals.html). These funders paid for material costs for the study. Boehringer-Ingelheim Ltd. provided the non-steroidal anti-inflammatory pain medication meloxicam (Metacam®) (SM). The funders had no role in study design, data collection and analysis, decision to publish, or preparation of the manuscript.

**Competing interests:** The authors have read the journal's policy and have the following competing interests: Boehringer-Ingelheim Ltd. provided the non-steroidal anti-inflammatory pain medication meloxicam (Metacam®). This does not alter our adherence to PLOS ONE policies on sharing data and materials. There are no other patents, products in development or marketed products associated with this research to declare.

MCP-1 concentrations ($P < 0.001$). Correlations between MCP-1 concentrations and synovial histopathologic grading and its discriminative ability suggest its potential as a synovitis biomarker in canine stifle OA associated with CrCL rupture.

## Introduction

The most common cause of stifle (knee) osteoarthritis (OA) development in dogs is degenerative (non-traumatic) cranial cruciate ligament rupture (CrCLR) [1, 2]. The resulting morbidity arising from the joint instability and related OA has an estimated annual treatment cost of $1.32 billion in the United States [3]. Dogs that develop OA associated with degenerative CrCLR have reported incidence of bilateral CrCLR that ranges from 18–61.3% [4, 5]. In cases of unilateral degenerative CrCLR, subsequent CrCLR in the contralateral stifle is often reported approximately a year after the initial diagnosis with the reported risk ranging from 22–54% [4, 6]. Despite the focused and extensive research on stifle OA in dogs, none of the therapeutic and management strategies have proven efficacious beyond alleviating symptoms and do not control the progression of OA in this joint [7]. Factors considered to be contributing to the limited success include the complexity of the etiology and pathophysiology of canine OA, and a lack of robustly validated biomarkers of OA suitable as reliable outcome measures in all stages of the disease [8]. Clinical examination and standard digital radiography are unable to detect early canine pre-clinical OA changes or objectively evaluate responses to interventions. Therefore, the assessment of molecular changes in biological fluids (i.e., serum, synovial fluid and urine) as soluble (wet) biomarkers of the disease is an area of interest in OA-related research [9, 10]. Additional challenges in soluble biomarker research are posed by the complexity of the sources of the molecules associated with the disease, and the presence of cross-talk among joints that has made identification of a single representative biomarker for identifying and evaluating OA unrealistic [11, 12]. The process of validating biomarkers in the clinical setting also remains a challenge due to difficulties in case definition and selection and in the recruitment and associated costs of verifying the repeatability and accuracy of candidate biomarkers [13, 14]. The investigated biomarkers of canine stifle OA include pro-inflammatory mediators (e.g., cytokines), degradative enzymes and their inhibitors (e.g., matrix metalloproteinases), and extracellular matrix proteins (e.g., proteoglycans, collagen type II degradation or synthesis products) and their composites [8, 15]. Evaluations of candidate biomarkers for canine stifle OA have been performed using meniscectomy, cranial cruciate ligament transection (CrCLt), and groove models [16–19]. However, the naturally-occuring OA secondary to CrCLR model is of particular interest due to its frequent occurrence in the clinical setting, and similarities to human knee OA [10, 20, 21]. A study by Garner *et al.* (2011) investigated multiple candidate diagnostic OA biomarkers in serum, synovial fluid (SF), and urine based on experimentally-induced (i.e., CrCLt), and clinical cases of stifle OA associated with naturally occurring canine CrCLR [15]. These authors suggested two matrix metalloproteases (MMP-2 and MMP-3) as candidate serum biomarkers, and a panel of three SF biomarkers (keratinocyte-derived chemoattractant (KC), interleukin 8 (IL-8) and monocyte chemoattractant protein 1 (MCP-1)) as diagnostic biomarkers of stifle OA [15]. However, the number of clinical cases in that study was limited (n = 10), and to the authors' knowledge, no further studies evaluating these panels of serum and SF biomarkers have been published.

The overarching goal of this study was to evaluate MMP-2 and MMP-3 in serum, and KC, IL-8 and MCP-1 in SF as stifle OA biomarkers in dogs. We hypothesized that MMP-2 and MMP-3 in serum and KC, IL-8 and MCP-1 in SF would have discriminative abilities as

diagnostic, monitoring and predictive biomarkers of OA with close associations with concurrent macroscopic and microscopic stifle joint pathologies related to naturally-occuring OA associated with CrCLR in dogs. The first objective of this study was to evaluate the discriminative ability of these candidate biomarkers for canine stifle OA associated with naturally occurring CrCLR. The second was to evaluate temporal changes in these serum and SF biomarker concentrations after surgical stabilization of the CrCL deficient stifles. The third objective was to assess these serum and synovial biomarkers in evaluating changes in the contralateral stifles that were stable at the time of initial enrollment of the OA group in the study to evaluate the possibility of predicting the fate of the CrCL in the contralateral stifle. The fourth objective was to assess the biomarker associations between additional joint pathologies, including the presence of meniscal injury, degree of CrCL tear, and histological grade of synovitis in the OA affected stifles.

## Materials and methods

In accordance with the Guide to the Care and Use of Experimental Animals of the Canadian Council on Animal Care (#11–062), the Animal Care Committee of the University of Prince Edward Island approved this prospective clinical cohort observational study. Owners of clinical cases and control dogs were required to complete and sign a written consent form prior to enrollment of dogs in the study. Sample size calculation for this study was based on a 95% confidence level and power range 95% for an unmatched case-control study with a single control per two OA cases, resulting in a group size estimate of 27 OA cases with an effect size of 1.287 (for MMP-2 and MMP-3) based on an estimated 80% prevalance of OA. Because there are no true prevalence data for stifle OA in dogs, group sizes were increased based on previous studies [22] to allow for potential sample losses and patient variations (~50 OA and 50 control dogs). An additional 10% allowance was made for the exclusion of samples not found to meet inclusion criteria. This increased sample size was also necessary to allow a meaningful number of the OA group dogs (~45–50) to be included in the recheck groups given the propensity for clinical cases to be lost to follow up.

### Animals

Adult, medium to large breed (>15kg bodyweight) client-owned dogs with a clinical diagnosis of naturally occurring CrCLR in one or both stifles were recruited in the OA group. Additional inclusion criteria for the OA group included being free of systemic disease based on complete physical, neurologic and orthopedic examinations, and lack of significant abnormalities on a complete blood count and serum biochemistry profile. Exclusion criteria were additional orthopedic abnormalities (e.g., patella luxation), history of surgery or use of systemic corticosteroids within 4 weeks before recruitment, history of traumatic CrCL tear, or a history of intra-articular corticosteroid injections. Pre-surgical diagnosis of CrCLR was based on observation of clinical lameness in the affected hind limb and the presence of one or more of the following criteria: pain on hyperextension of the stifle, palpable joint effusion, positive cranial drawer test, or positive tibial thrust test [6]. A diagnosis of stifle OA was confirmed based on orthogonal radiographs of the stifle joint and intraoperative observation of CrCLR with gross evidence of OA (e.g., osteophytes, synovitis, joint effusion, cartilage lesions) [23].

The unmatched control group included adult, medium to large breed (>15kg bodyweight) dogs euthanized for reasons unrelated to this project. These dogs were free of orthopedic or systemic abnormalities based on physical and orthopedic examinations immediately prior to euthanasia. Postmortem examination of both stifles was performed to confirm the lack of gross abnormalities in all compartments of both joints. Dogs with concurrent systemic illness,

or positive for dirofilariasis, ehrlichiosis, anaplasmosis, or borreliosis based on a commercial ELISA-based test (SNAP® 4D® Test, IDEXX Laboratories, Westbrook, ME.) were excluded.

All OA group dogs with CrCLR underwent either tibial plateau leveling osteotomy or lateral fabellotibial suture techniques on the CrCL deficient joints to stabilize the stifle [24, 25]. For bilaterally affected dogs, single-stage bilateral surgeries or staged procedures were performed based upon surgeon preference or the owner's financial constraints. In the case of the staged procedures, the second stifle was not operated until after the conclusion of the study; these dogs were not recruited a second time within the study. At the time of surgery during exploration of the joint via arthroscopy or arthrotomy, the CrCL of the operated stifle was evaluated for the presence of a partial or a complete tear of the ligament. The medial and lateral menisci in the operated stifle were also evaluated for the presence or absence of any tear or damage at the time of surgery. In all operated stifles, damaged or diseased components of the CrCL and menisci were debrided with no attempts at reconstruction of the torn ligaments or menisci. The postoperative pain management regimen included; hydromorphone (Hydromorphone hydrochloride, 2mg/ml, Sandoz Canada Inc., Quebec) at 0.05–0.1mg/kg IV or SC every 4–6 hours for 48 hours, meloxicam (Metacam®, 1.5mg/ml, Boehringer-Ingelheim, Burlington, ON) at 0.1mg/kg, orally, every 24 hours for 7–10 days, and tramadol (Tramadol, Chiron, Compounding Pharmacy Inc., Guelph, ON) at 4-8mg/kg, orally, every 8–12 hours for 5–7 days.

## Sample collection

**Serum samples.**   At the initial visit (T1), a 4 ml venous blood sample was collected preoperatively and allowed to clot before being centrifuged at 5000 rpm (3400 g) for 5 minutes. The serum was separated and stored in cryovials (Nalgene Cryogenic tubes, VWR International, Batavia, IL, USA) at -80˚C for later batch analysis. A subset of the OA dogs were selected for re-evaluation and sampling at 4 (T2), and 12 (T3) weeks after surgery. At each revisit (T2 and T3), physical and orthopedic examinations were performed prior to radiographic imaging of stifles under sedation. Venous blood collection and processing of samples were performed using the same methodology described for the initial visit. Exclusion criteria for this subset of OA dogs included clinical or radiographic evidence of infection at the level of the joint or implant site, catastrophic implant failure, or diagnosis of any other systemic illness. Blood samples from control dogs were obtained once before euthanasia and similarly processed.

**Synovial fluid samples.**   Samples were collected from both stifles irrespective of whether one or both stifles were affected with CrCLR at the T1 time point. On the day of surgery, the dogs in the OA group were placed under general anesthesia, and stifles were clipped free of hair and prepared for aseptic arthrocentesis. Sterile 6 ml syringes with 1.5", 22-gauge hypodermic needles were used to aseptically aspirate SF from each joint. The SF samples were placed in labeled cryovials and frozen immediately at -80˚C for later batch analysis. Surgical stabilization of the CrCL deficient stifle(s) was performed after the SF sample was obtained. At the subsequent visits (T2 and T3), bilateral stifle SF sample collections were performed using aseptic technique and under sedation following radiographs of the joint. Samples obtained from the CrCL deficient stifle with OA were labeled as OA stifle (index stifle), and the samples collected from the opposite stifle that had an intact CrCLR (stable stifle), based on clinical examination, at the time of enrollment and throughout the study were labeled as contralateral. Therefore, a dog that had bilateral CrCL deficient stifles, did not have a contralateral sample. In the control group of dogs, the SF samples from both stifle joints were aseptically collected immediately after euthanasia using the same method and similarly stored. Both SF samples obtained from each control dog were labeled as control samples. A flow diagram in Fig 1 summarises the sampling order from the OA and control group dogs.

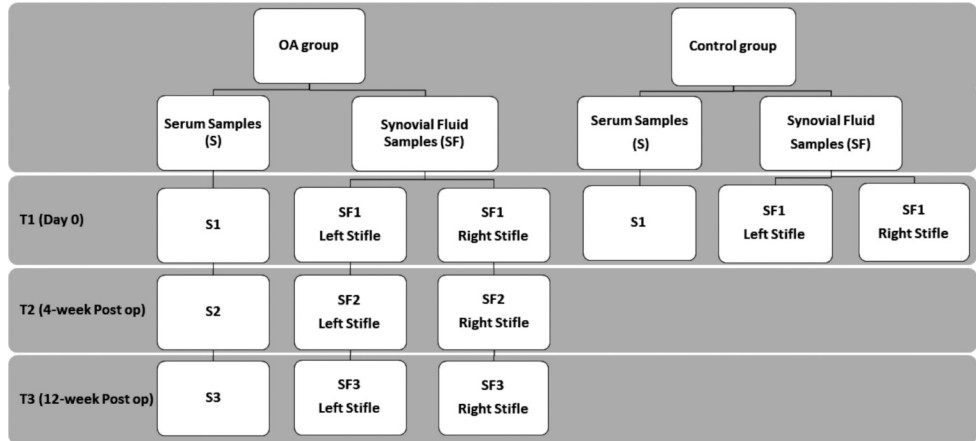

**Fig 1. Flow diagram of serum (S) and synovial fluid (SF) sampling from control and osteoarthritis (OA) groups.**
Samples from the OA group were obtained at three time points and only once from the control group. Samples
obtained from stifles in control dogs were labeled as controls while samples from right and left stifles of OA group were
labeled as index stifle or contralateral stifle depending on whether the cranial cruciat ligament was intact or not. Only a
subset of the dogs in the OA group were recruited for sampling at T2 (4-week post op) and T3 (12-week post op) time
points.

**Multiplex bead assay.** The serum and SF samples frozen at -80˚C were shipped overnight
on dry ice to the University of Missouri's Comparative Orthopedic Laboratory and stored at
-80˚C. At the time of analyses, samples were thawed, and an aliquot from each SF (100μl), and
serum (50μl) sample was processed for testing. The SF samples were centrifuged at 14,000 rpm
for 10 minutes to pellet debris, and the supernatant removed. The SF was then incubated with
hyaluronidase (MPBiomedicals, LLC, Solon, Ohio) at 37˚C for 90 minutes to decrease viscos-
ity. After digestion, the SF samples were analyzed using a custom luminex canine cytokine
\chemokine bead panel for IL-8, KC, and MCP-1 (Millipore Corp. St. Louis, MO). Serum sam-
ples (50μl) were analyzed using custom human luminex performance MMP panel (R&D Sys-
tems, Minneapolis, MN) for MMP-2 and MMP-3 shown previously to cross-react with
samples of canine origin [26]. Samples were processed according to the manufacturer's proto-
col, and protein concentrations in the samples were determined using a Luminex 100/200 sys-
tem (Qiagen Inc., Valencia, CA) with settings according to the assay manufacturer's protocol
for each assay.

The luminex (xMAP assay) works by mixing the sample with small (5.6 micron) polysty-
rene microspheres. Each microsphere has a specific fluorescent signature that can be detected
and differentiated using the detector (Luminex™ 200TM, Luminex Corporation, Austin
Texas). Further, each microsphere type is charged with a monoclonal antibody for a specific
protein. Therefore, each set of microspheres is a specific test for the desired protein, and one
aliquot of a sample is used for testing multiple proteins by mixing the microspheres for differ-
ent proteins together with the sample, allowing binding to the microspheres. The amount of
protein bound to the beads is proportional to the amount of protein in the sample. Following
overnight incubation at 4˚C, the beads are washed and then mixed with a biotinylated poly-
clonal secondary antibody for each protein tested. The samples are then mixed with streptavi-
din-phycoerythrin, which binds to the biotin secondary antibody and places a fluorescent tag
on the bead proportional to the amount of protein bound to the bead. Using a luminex detec-
tor each bead is identified based on the fluorescent signature of the bead, and then the level of
phycoerythrin fluorescence is measured. A total of 40 beads for each protein species were

measured and the median fluorescence intensity for each is used to determine the concentration of each protein for each sample.

**Synovial biopsy.** In the OA group at the initial visit, the operated CrCLR stifle joints were approached via a standard medial arthrotomy [27] for surgical stabilization. The same surgical approach was used for both stifles of the control group dogs immediately after euthanasia. For each joint, a 1 x 2 cm piece of the synovial membrane was excised from the edge of the medial arthrotomy incision. The sections of the collected synovial tissues were fixed in 10% buffered formalin, embedded in paraffin, sectioned at 5 μm, and stained with hematoxylin and eosin (H&E). All sections were examined to rule out the presence of non-OA related changes (e.g., neoplasia, pyogranulomatous inflammation or evidence of infectious organisms), and then scored using a published synovitis grading system [28] by a blinded board certified veterinary pathologist (SAM). The samples were scored for the following parameters: synoviocyte changes (proliferation and hypertrophy), inflammatory infiltrates (neutrophilic infiltrates, fibrinous exudate, lymphoplasmacytic infiltrates, and lymphoid aggregates/follicles), and changes in the synovial stroma (villous hyperplasia, proliferation of fibroblasts, proliferation of blood vessels, cartilage/bone detritus, and hemosiderosis). For lymphoid follicles, cartilage/ bone detritus, and hemosiderosis a score of 0 to 2 was assigned; the other parameters were scored from 0–3 [28]. This grading system was chosen over other grading systems, since other systems have not been validated for clinical use in dogs, and lack details in histological characteristics that were deemed relevant in this study [29].

## Statistical analysis

Data distributions are presented in box plots with corresponding median and range reported. Natural logarithmic transformation was applied to the right-skewed measures before hypothesis testing. When evaluating the differences between the control and OA group samples, only the T1 samples from OA group were used. Additional comparisons between control and OA samples at subsequent visits were performed separately. General linear regression was used to compare serum biomarker concentrations between control and OA groups while adjusting for significant covariates. For evaluating individual SF biomarkers between OA affected stifles (index stifle), and contralateral stifles, and control stifles, mixed linear regression was used to account for dependency between the bilateral SF samples. Posthoc pairwise comparisons with Bonferroni adjustments were performed, and statistical significance was set as $P < 0.05$.

Receiver operating characteristic (ROC) analyses were performed to investigate the overall discriminative ability of individual biomarkers using area under the ROC curve (AUC). In addition, three logistic regression models were fit to the data to investigate the discriminative performance of combining different biomarkers: 1) MMP-2 and MMP-3, 2) IL-8, KC, and MCP-1, 3) all investigated biomarkers. Predicted probabilities derived from these logistic regression models were then evaluated using the ROC analyses. For the biomarkers (or their combinations) that yielded an AUC $\geq 0.90$, the optimal cutoff value based on the Youden's index was determined, and corresponding sensitivity and specificity were reported.

To evaluate temporal changes in biomarker concentrations, longitudinal data on the biomarker concentrations and contralateral stifles were analyzed. Mixed regression was used to compare biomarker concentrations across the three sampling time points. To evaluate the ability of serum and synovial biomarkers as predictors of the fate of contralateral CrCLR (i.e., time to rupture), Kaplan-Meier survival analysis was performed in the subject of the OA dogs with unilateral CrCLR. To run Kaplan-Meier survival analyses, each biomarker was dichotomized into high and low groups using the optimal cutoff value determined by ROC analysis or the

median concentration. Log-rank tests were then performed to compare the survival curves between high and low groups.

Finally, mixed linear regression and ROC analyses were performed to evaluate synovial histology grades. To assess correlations between synovial histology grades and the serum and SF biomarkers, Spearman's correlation coefficient was used. To evaluate the ability the histological grading system to predict class labels, the bilateral cases were excluded to avoid repeated measures, and the optimal cut off points for distinguishing between OA and control dogs for sum of inflammatory infiltrate grades, sum of synovial stroma grades, and sum of all scores were determined.

## Results

Sixty-two dogs in the OA group and 50 dogs in the control dogs were included in the study. The median (range) for weight in the OA and control groups were 37.7 (18.3–81.0) kg and 24.1 (15–55.5) kg, respectively. The median (range) for age in the OA and control groups were 4.9 (1.3–10.9) years and 2 (0.5–8) years old, respectively. Weight and age were significantly different between the OA and control groups and therefore were adjusted for in all the group comparisons. There were two intact males, two intact females, 25 neutered females and 33 neutered males in the OA group. In the control group, there were two each of neutered males and neutered females, 20 intact females, and 26 intact males. The sex distribution was not different between the groups ($P = 0.962$, Pearson's chi-squared test) and was not further considered in analyses. The most common dog breed in the OA group was Labrador Retriever (n = 22) and in control group, mixed breeds (n = 28) (Table 1).

There were 50 control dog serum samples. For the OA dogs there were 62, 47 and 46 serum samples available from T1, T2, and T3 sampling time points respectively. There were 93 control dog SF samples. For the OA dogs there were 118, 92 and 90 SF samples available from T1, T2, and T3 sampling time points respectively. The median time reported by clients from initial

**Table 1. Frequency of dog breeds.**

| OA Group (n = 62) | | Control Group (n = 50) | |
|---|---|---|---|
| **Dog Breed** | **Count** | **Dog Breed** | **Count** |
| Labrador Retriever | 22 | Mixed breed | 28 |
| Golden Retriever | 7 | Pit Bull Terrier | 11 |
| Mastiff | 5 | German Shepherd | 4 |
| Boxer | 4 | Labrador Retriever | 3 |
| Mixed Breed | 6 | Australian Cattle Dog | 2 |
| Newfoundland Dog | 3 | Pointer | 1 |
| Rottweiler | 3 | Rottweiler | 1 |
| Bernese Mountain Dog | 2 | | |
| German Shepherd | 2 | | |
| Valley Bulldog | 2 | | |
| Airedale Terrier | 1 | | |
| Chesapeake Bay Retriever | 1 | | |
| Doberman Pinscher | 1 | | |
| Great Dane | 1 | | |
| Leonberger | 1 | | |
| Staffordshire Terrier | 1 | | |

The breed frequencies are listed separately for osteoarthritis (OA) group and control group dogs.

clinical signs of CrCLR to the time of initial presentation (chronicity of disease) was 92 days (1–732) and this information was available for 54 of the 62 dogs in the OA group. There were 39 complete and 31 partial CrCL tears observed in the operated stifles. When the effect of the presence of complete versus partial tear of the CrCL on the biomarker levels was evaluated, the presence of a complete CrCLR resulted in a statistically significant increase in KC concentration in the joint *(P = 0.015)* but not in other biomarkers. There were 38 medial meniscal tears in the 70 operated stifles. None of the operated stifles had any lateral meniscal tears. When the intraoperative status of the meniscus (intact versus torn) alone or in combination with CrCLR was evaluated, no significant biomarker concentration correlations were identified.

## Serum biomarkers

The distributions of MMP-2 and MMP-3 concentrations are presented in S1 Fig. The mean MMP-2 concentration was 2.4 fold higher in the OA group compared to the control after adjusting for the age and weight of dogs; indicating a large but statistically non-significant difference *(P = 0.074)*. The mean MMP-3 concentration, after adjustment for age and weight of dogs, was 5% lower in the OA group compared to the control group; indicating a small but statistically non-significant difference *(P = 0.796)*.

The AUC for MMP-2 and MMP-3 was 0.61 (95% CI 0.50–0.71) and 0.53 (95% CI 0.42–0.64), respectively demonstrating poor overall discriminative abilities. Combining the MMP-2 and MMP-3 did not improve the discriminative performance (Table 2). The serum concentrations of MMP-2 and MMP-3 of control dogs were compared to the two recheck time points (T2 and T3). There were no significant differences for MMP-2 levels between control and OA group recheck time points T2 and T3 *(P = 0.058 and P = 0.069 respectively)*, and for MMP-3 levels at either time points *(P = 0.681 and P = 0.951 respectively)*. When bilateral versus unilateral stifle OA was accounted for within the analyses, the discriminatory performance of both biomarkers did not improve (values not reported). When changes over time for MMP-2 and MMP-3 biomarkers in the OA group were evaluated (S2 Fig), the only statistically significant difference was between the T1 and T2 time points for MMP-2 concentrations *(P <0.001)*. The discriminative ability of serum biomarkers at both T2 and T3 time point measurements remained low (Table 2).

## Synovial fluid biomarkers

Comparison of the concentrations of the SF biomarkers at T1 from the OA group (index and contralateral stifles) and the control dog stifles showed significant differences between the

**Table 2. Discriminative performance of serum MMP-2 and MMP-3.**

| Serum Biomarker | Comparison | AUC | 95% CI |
|---|---|---|---|
| | T1 vs Control | 0.61 | 0.50–0.71 |
| MMP-2 | T2 vs Control | 0.69 | 0.58–0.79 |
| | T3 vs Control | 0.67 | 0.56–0.78 |
| | T1 vs Control | 0.53 | 0.42–0.64 |
| MMP-3 | T2 vs Control | 0.54 | 0.42–0.66 |
| | T3 vs Control | 0.58 | 0.46–0.70 |
| MMP-2 and MMP-3 | T1 vs Control | 0.62 | 0.53–0.71 |

The overall performance of serum MMP-2 and MMP-3 at different measurement time points (T1, T2, and T3) and combined on discriminating between dogs with and without osteoarthritis. The discriminative performance is measured using area under the ROC curve (AUC). T1: initial visit, T2: 4-week recheck, T3: 12-week recheck time points.

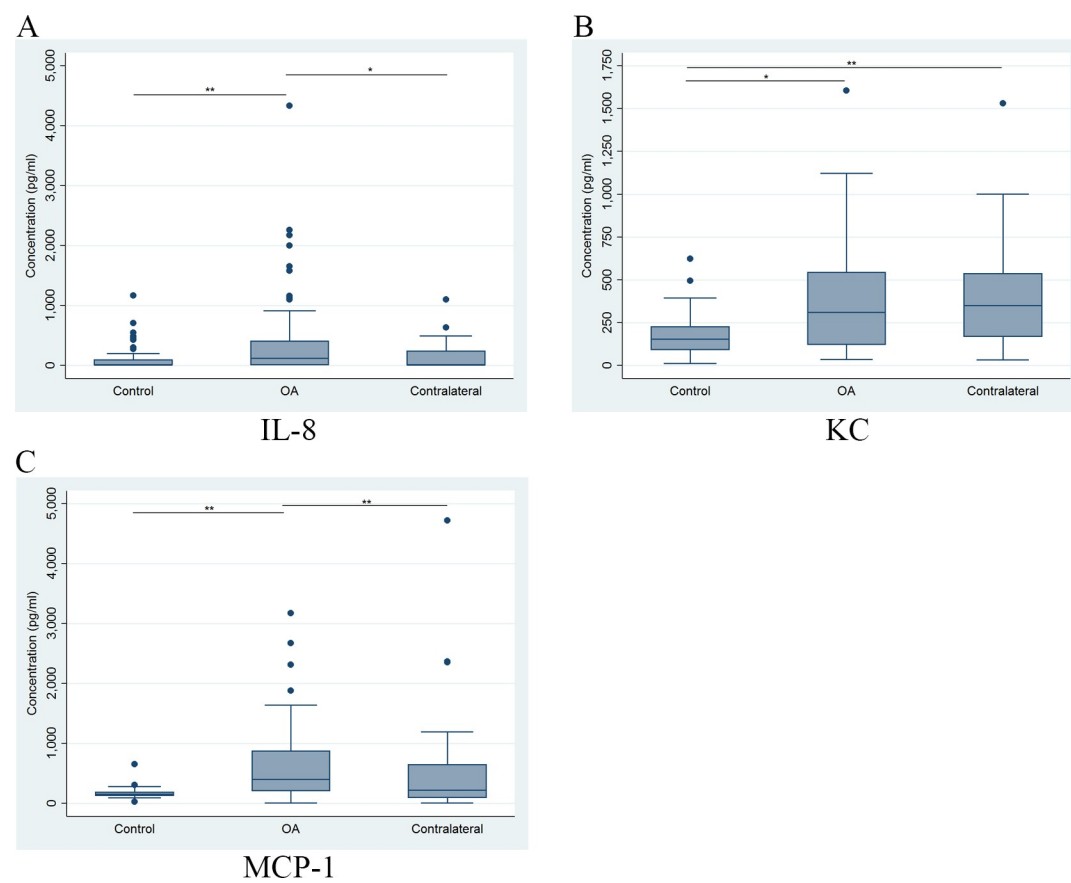

**Fig 2. Comparison of synovial fluid biomarkers (IL-8, KC and MCP-1) between control and OA group samples.**
Comparison of the distribution of IL-8 (A), KC (B), and MCP-1 (C) synovial fluid concentrations (pg/ml) at the initial visit
(T1) for stifles with OA (OA), the stable stifles in the OA dogs (contralateral) and control dogs (control). The horizontal line
inside each box is the median and the upper and lower edges of box present the inter-quartile range (IQR). The whiskers are
either $1.5 \times$ IQR or the range, whichever is smaller. Dots outside the fence are outliers. *Statistically significant if $P < 0.05$, **
or if $P < 0.01$.

index stifle and control stifles for all three biomarkers (Fig 2). IL-8 and MCP-1 levels were sig-
nificantly different between the index and contralateral stifles, while KC levels in the contralat-
eral stifle were significantly different from the control but not the index stifle samples (Fig 2).
A summary of comparisons between concentrations of the three SF biomarkers among con-
trols, the index OA stifles and contralateral stifles at all three time points are presented in
Table 3. Based on these results, there were significant differences between T1 and T2 for SF
concentration of all three biomarkers between the index stifle and control samples. There were
significant differences between SF concentrations of index and contralateral stifles in the OA
group for IL-8 at T1 and T2 and for MCP-1 at all three time points. However, the only differ-
ences observed between control versus contralateral stifles was for KC concentrations at all
three time points.

Comparisons of SF biomarker concentrations at different time points in either index or
contralateral stifles in the OA group are shown in Fig 3. For the OA stifles, there was a signifi-
cant decrease in all three biomarker concentrations at T3 compared to T1 and T2. However,
there was no significant difference in biomarker concentrations between T1 and T2 despite a
trend for elevation in all three biomarkers in the OA stifles. The results showed no significant
change over time in the contralateral stifles (Fig 3).

**Table 3. Synovial fluid biomarker differences between among groups and over time.**

| Synovial Biomarker | Comparisons | T1 (*P* value) | T2 (*P* value) | T3 (*P* value) |
|---|---|---|---|---|
| **IL-8** | OA vs Control | 0.001* | 0.003* | 0.411 |
| | OA vs Contralateral | 0.010* | 0.007* | 0.745 |
| | Control vs Contralateral | 0.405 | 0.658 | 1 |
| **KC** | OA vs Control | 0.010* | <0.001* | 0.096 |
| | OA vs Contralateral | 0.872 | 0.978 | 1 |
| | Control vs Contralateral | 0.001* | <0.001* | 0.043* |
| **MCP-1** | OA vs Control | <0.001* | <0.001* | 0.295 |
| | OA vs Contralateral | <0.001* | <0.001* | 0.043* |
| | Control vs Contralateral | 1 | 1 | 1 |

Mixed linear regression model adjusted for age and weight followed by post hoc pairwise comparison with Bonferroni correction at each time point.

*Statistical significance ($P < 0.05$). The comparison is for control dogs (both stifles sampled), OA group with index OA stifle and the contralateral unaffected stifle.

For the ROC analysis, the SF biomarkers (IL-8, KC and MCP-1) were compared between the right and left stifles in control dogs. As no statistically significant difference was found, the mean concentration of each biomarker for the right and left stifle of each control dog was used in the ROC analysis when comparing control dogs with OA groups (i.e., index and contralateral stifle values). When these SF biomarkers were evaluated for their discriminative abilities, the MCP-1 was the only biomarker that showed an acceptable performance with an AUC of 0.91 (95% CI: 0.83–0.98) distinguishing between index stifles and controls (Fig 4). The optimal cutoff value based on the Youden's index was determined at a MCP-1 concentration >265 pg/ml. This cutoff value resulted in a sensitivity and specificity of 85% (95% CI: 72–94) and 98%

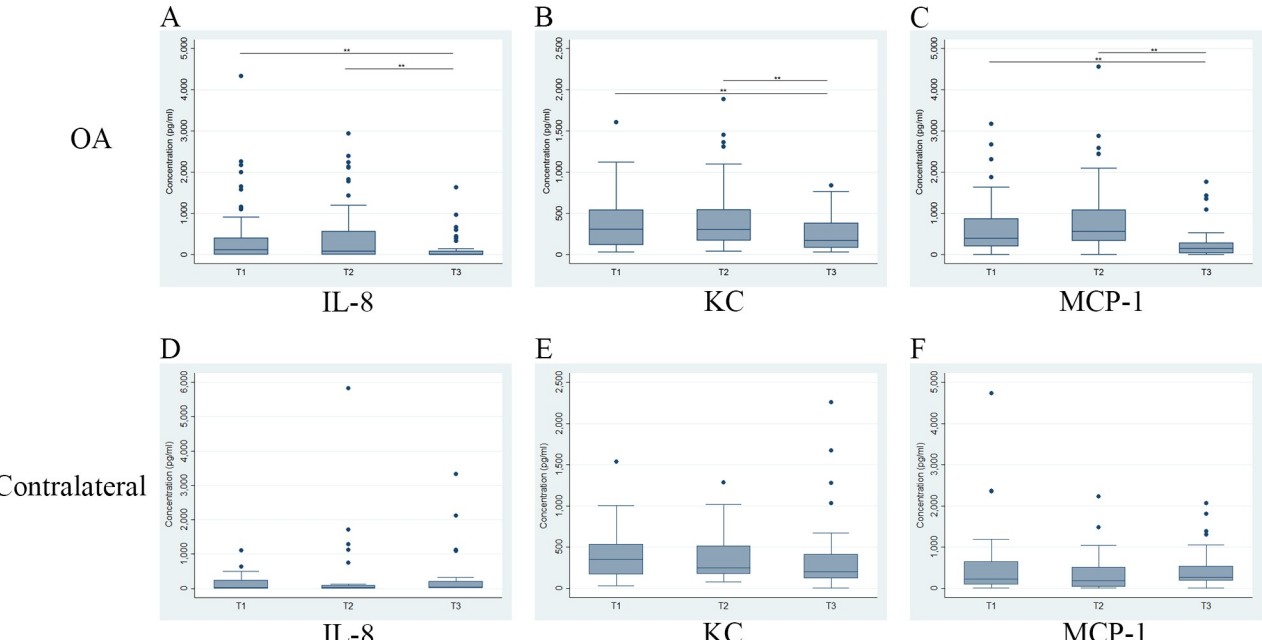

**Fig 3. Comparison of temporal changes of concentrations of synovial fluid IL-8, KC and MCP-1 biomarkers for the OA group dogs.** The first row of graphs are for synovial fluid samples from index stifles (OA) and the second row are for synovial fluid samples from the stable stifles of dogs in the OA group (contralateral). The time points evaluated are initial visit (T1), 4-week recheck (T2) and 12-week recheck (T3). The horizontal line inside each box is the median and the upper and lower edges of box present the inter-quartile range (IQR). The whiskers are either $1.5 \times$ IQR or the range, whichever is smaller. Dots outside the fence are outliers. * if $P < 0.05$; ** if $P < 0.01$.

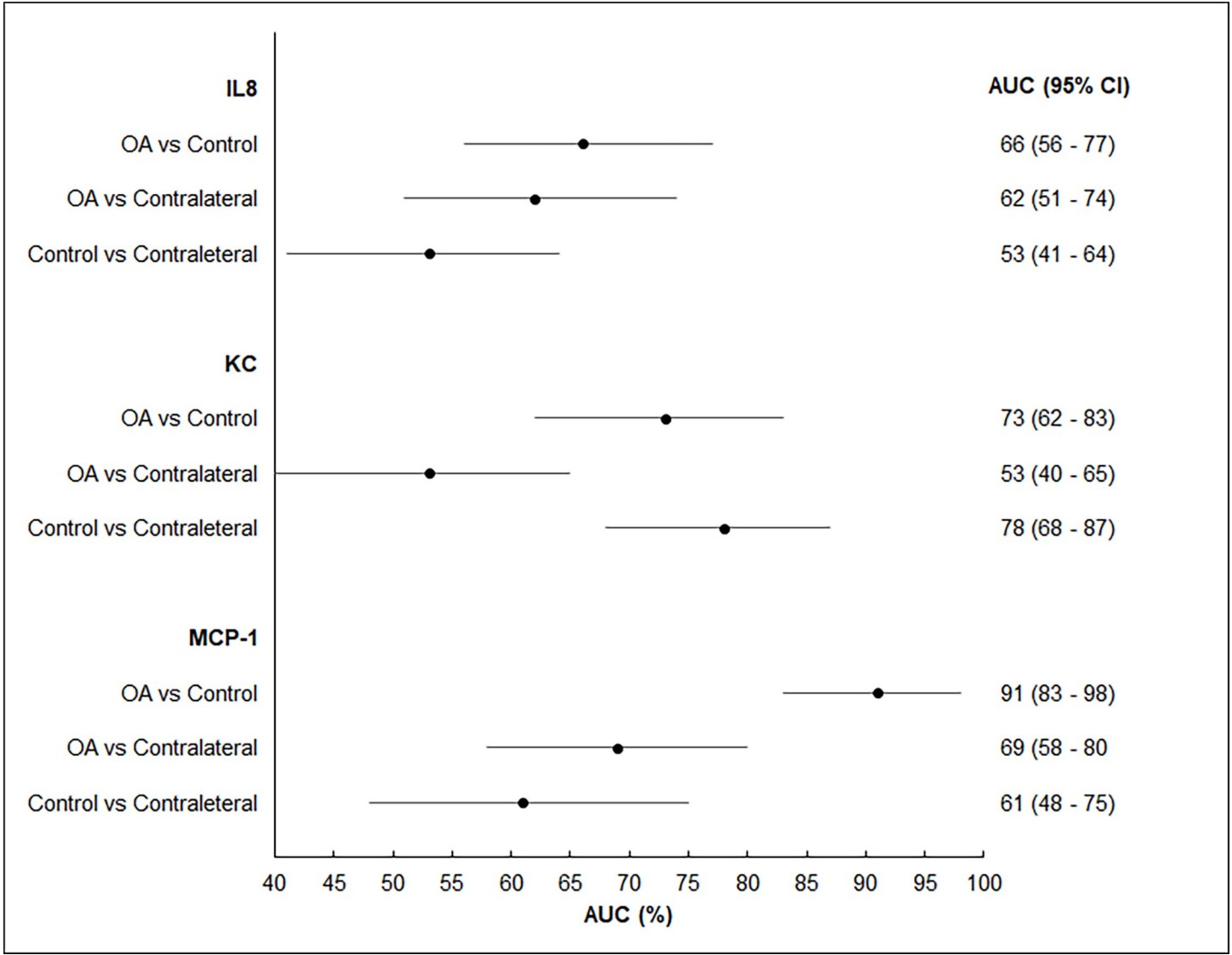

**Fig 4. Discriminative performance of synovial fluid IL-8, KC and MCP-1 biomarkers' ROC model performance at T1 time point for the OA group compared to control.** The forest plot depicts the area under the curve (AUC) and 95% confidence interval (CI) to quantify the overall discriminative performance. Comparisons are between synovial fluid samples from the index (OA) and contralateral stifles from the OA group and control synovial fluid samples. An AUC = 1 indicates a perfect discriminative ability, whereas an AUC = 0.5 indicates no discriminative ability. It desired to have an AUC ≥ 0.9 for a clinically acceptable performance.

(95% CI: 89–100), respectively. The same methodology was used to compare the control biomarker values with the index and contralateral stifle samples from the T2 and T3 visits. The MCP-1 model continued to have a superior discriminative performance among the investigated biomarkers. At T2, the statistically optimal cutoff value was at > 265 pg/ml with a sensitivity and specificity of 91% (95% CI: 77–98) and 98% (95% CI: 89–100) respectively. However, at T3 none of the biomarkers had an AUC > 0.90.

Follow up data based on post-surgery follow-up visits and telephone survey was available for 40 dogs for an average of 850 (104–2692) days. Of the 40 dogs, 17 (42.5%) developed a subsequent CrCLR in the contralateral stifle that had been stable at the time of inclusion in the study. The cutoff point selected for each biomarker's concentrations in the two groups was the median concentration of the biomarker evaluated, except for MCP-1, where the optimal cut-

off value of 265 pg/ml (obtained from the predictive model) was utilized. However, none of the biomarker concentrations (SF or serum) were shown to have discriminative predictive ability for time to development of contralateral CrCLR.

## Synovial histopathology

Synovial tissue samples from both stifles for histopathological grading were available in all 50 control dogs (n = 100, left and right stifles). There were 62 OA dogs, six with bilateral CrCLR that had surgery and both stifles sampled (n = 12, left and right stifles), and 56 dogs that had unilateral surgery performed. Six of the 56 unilateral dogs did not have adequate tissue available for histopathological grading; the remaining 50 had histopathological grading for their surgical biopsies (n = 50; 26 right and 24 left stifles). Therefore, 62 synovial tissue samples of the OA group were available for grading. There was no statistically significant difference between the synovial histopathology grade between the right and left stifle in the control dogs; therefore the mean histopathology grade of right and left samples from each control dog was used. The comparison of histopathological grades between the OA and control groups using mixed regression model revealed statistically significant differences ($P < 0.001$) between the sum of synoviocyte scores, sum of inflammatory infiltrate scores, sum of synovial stroma grades, and sum of all scores. The optimal cutoff points for each grade in determining OA versus control were calculated and are presented in Table 4.

To evaluate correlations between histopathological grades and the serum and synovial biomarkers, only dogs (n = 91) with both biomarker and histopathological grades available were included in the analysis. In the OA group, only unilateral cases were included (n = 41). The correlations between histology grades and serum biomarkers MMP-2 and MMP-3 were all weak (Spearman's rho < 0.2). For SF biomarkers, there were multiple significant correlations between the biomarkers and the histopathological grades (Table 5). The sum of inflammatory infiltrate score was significantly correlated with all three SF biomarkers ($P < 0.01$), while the sum of synovial stroma score and the sum of all scores were significantly correlated with IL-8 and MCP-1 only ($P < 0.003$), and the sum of synoviocyte score was only significantly correlated with MCP-1 ($P < 0.001$).

## Discussion

Based on the findings in this study, our hypothesis was partially rejected as the concentrations of the two serum biomarkers (i.e. MMP-2 and MMP-3) were unable to discriminate between the OA and control dogs (i.e., AUC = 0.5 for both biomarkers). MMP-2 (i.e., gelatinase A, 72KDa type IV collagenase) is produced from various tissues in the synovial joint, and is involved in the extracellular matrix degradation process in both articular cartilage and subchondral bone [30]. In human OA, serum MMP-2 levels have been shown to be elevated compared to controls [31], which correlates with the observed trend in the current study although it was not statistically significant. Stifle SF levels of MMP-2 and MMP-3 have been previously

**Table 4. Optimal cut-off points, based on the Younden's index, for summed scores in each category of histopathological grading of synovitis between OA and control dogs.**

| Histopathological Scores | Optimal cutoff | Sn (95%CI) | Sp (95% CI) |
|---|---|---|---|
| Sum of inflammatory infiltrate scores | > 2 | 93% (80–99) | 91% (84–96) |
| Sum of synovial stroma scores | > 5 | 98% (87–100) | 94% (87–98) |
| Sum of all scores | > 13 | 98% (87–100) | 97% (92–99) |

Sensitivity (Sn) and specificity (Sp) of each cut-off point is reported with the 95% confidence interval (CI).

**Table 5. Correlation between synovial fluid biomarkers (i.e., IL-8, KC, MCP-1) and histopathological grade of biopsied synovial membrane.**

| Synovial membrane histological grading | Synovial fluid biomarkers | | |
|---|---|---|---|
| | **IL-8** | **KC** | **MCP-1** |
| | **CC (P value)** | **CC (P value)** | **CC (P value)** |
| **Synoviocyte** | | | |
| •Proliferation | 0.32 (0.007) | | 0.38 (<0.001) |
| •Hypertrophy | | | 0.35 (<0.001 |
| **Sum of Synoviocyte scores** | | | 0.38 (<0.001) |
| **Inflammatory infiltrate** | | | |
| •PMNs | | 0.31 (0.02) | 0.49 (<0.001) |
| •Fibrin | 0.32 (0.007) | | 0.55(<0.001) |
| •Lymphoplasmacytic infiltrate | 0.35 (0.002) | 0.31(0.017) | 0.55 (<0.001) |
| •Lymphoid aggregates/follicles) | | | 0.36 (0.001) |
| **Sum of Inflammatory infiltrate scores** | 0.34(0.004) | 0.32(0.011) | 0.56 (<0.001) |
| **Synovial Stroma** | | | |
| •Villous hyperplasia | | | 0.45(<0.001) |
| •Proliferative fibroblasts/fibrocytes | | | 0.45(<0.001) |
| •Proliferative blood vessels | 0.33 (0.000) | 0.32 (0.011) | 0.6(<0.001) |
| •Cartilage/bone detritus | | | |
| •Hemosidrosis | 0.34 (0.003) | | 0.50 (<0.001) |
| **Sum of Synovial Stroma scores** | 0.34 (0.003) | | 0.54 (<0.001) |
| **Sum of all scores** | 0.35 (0.001) | | 0.54 (<0.001) |

The P value and Spearman's correlation coefficient (CC) are provided with significance set at P < 0.05.

evaluated in clinical models of CrCLR in dogs and were significantly elevated in affected compared to control joints [32, 33]. An *in vitro* model has shown that intact CrCL explants can produce significantly higher MMP-2 and MMP-3 proteins, and this ability is reduced as more CrCL is damaged and as the process becomes more chronic [26]. However, the serum MMP-2 and MMP-3 levels reported by Garner *et al.* (2011) were significantly higher in normal dogs and dogs that had undergone stifle stabilization due to CrCLR (8 and 12 weeks after surgery) when compared to presurgical dogs with CrCLR [15]. However, their results were not corrected for body weight or age of the dogs [15]. In the current study, MMP-3 levels were initially higher in the OA group, but once adjustments for the age and weight of the dogs were made, the MMP-3 levels were 5% lower in the OA group but the difference was not statistically significant. The observed trend in MMP-3 levels in the OA group in this study is in agreement with the Garner *et al.* (2011) findings [15], and emphasizes the importance of accounting for variables such as age and weight of patients. Serum MMP-3 (stromelysin-1) is a key enzyme associated with cartilage degradation [34] and is elevated in human serum, plasma and tissue samples in acute destructive OA with acute pain of the hip joint [35]. The MMP-3 in the SF samples of dogs with rheumatoid arthritis is elevated [36] but gene expression of MMP-3 was not different in a canine CrCLR model compared to normal dogs [37]. This may be an indication that MMP-3 may warrant further investigation as a biomarker in differentiating rheumatoid arthritis and other forms of arthritis. These discrepancies between previous reports and the current study regarding serum MMP-2 and MMP-3 as OA biomarkers may be due to our larger sample size and variations in the severity of disease in our study population. The larger control group in our study may also have provided a wider natural variation in measured serum levels of these biomarkers.

Keratinocyte-derived chemoattractant (KC), also known as CXCL1 (C-X-C motif ligand 1), is a neutrophil chemoattractant in the same CXC chemokine subfamily as IL-8 with similarities to growth regulated oncogene-alpha (GRO∝) in humans [38]. It is known to be upregulated in chondrocytes of humans with OA and rheumatoid arthritis [39]. In dogs, KC as a stifle OA biomarker is less sensitive than MCP-1 and IL-8 due to the presence of overlap between normal and OA dogs [15]. In the current study, KC concentrations discriminated between control and OA joints as well as the contralateral joint. However, we were unable to demonstrate a difference between the KC levels of the OA and the contralateral stifle joints in the OA group. This KC elevation in the SF of stifles with OA associated with CrCLR in dogs was observed in the Garner *et al.* (2011) cohort. However, in that study, pre-surgery samples from the contralateral stifles had not been obtained [15]. The KC elevations in the contralateral stifles in this study may have been due to the presence of pre-existing synovitis in the contralateral stifle joint of dogs with unilateral degenerative CrCLR [40].

MCP-1 (i.e., CCL2) is of the C (γ) chemokine family and is considered a potent attractant of monocytes to sites of inflammation [41]. It has been widely investigated as a potential target for the treatment of diseases such as rheumatoid arthritis, atherosclerosis, and insulin-resistant diabetes [41]. IL-8 (a.k.a., CXCL-8, neutrophil activating peptide-1, NAP) has been implicated as a contributor to the pathophysiology of OA by promoting chondrocyte hypertrophy and apoptosis that ultimately results in cartilage degradation in the joint [42]. The SF IL-8 and MCP-1 concentrations in this study were significantly different between CrCLR stifle samples and the control, and between the CrCLR and the contralateral stifles. However, no significant differences in these biomarkers were noted between the control and contralateral samples. Increased synovial expression of IL-8 and MCP-1 as pro-inflammatory mediators has been documented in an experimental rabbit model of OA [43] and clinical rheumatoid arthritis in people [44]. IL-8 has also been associated with Borreliosis-associated arthritis [45]. Serum MCP-1 elevations in dogs have been previously associated with critical illness that is distinguishable from normal and postsurgical healthy patients [46] as well as in dogs with primary immune mediated hemolytic anemia [47]. When predictive ability of the SF and serum biomarkers in discriminating between control and OA group dogs were evaluated, MCP-1 in stifle SF was the only biomarker for which a cut off value (> 265 pg/ml considered OA) was calculated that predicted the class labels (OA versus control) with high sensitivity and specificity (> 90%). However, the predictive ability of MCP-1 was good only when comparing control dogs with OA dogs at T1 and T2 time points. This may have been due to moderation of the inflammatory response related to MCP-1 at the 12-week recheck time point.

When evaluating temporal changes in the biomarkers, MMP-2 was the only serum biomarker that showed a statistically significant difference between the T1 and T2 time points in OA dogs. All three SF biomarker levels in the OA stifles demonstrated a statistically significant decline between the initial visit and the 12-week recheck levels as well as between the 4-week and 12-week recheck levels. This finding may be an indication of reduced inflammatory response after surgical stabilization of the CrCL deficient stifles and the rest and more restricted activity imposed on these animals. Long-term studies of surgically stabilized stifles are warranted to evaluate the long-term changes in these biomarkers after the patient returns to normal daily activities and as the OA progresses in the joint. The lack of detectable differences between the initial visit and the 4-week recheck time point for all three SF biomarkers may have been due to the superimposed inflammation from the surgical intervention despite the postoperatively stable status of these joints. The incidence of subsequent CrCLR of the contralateral stifle in dogs that present with unilateral CrCLR is as high as 54%, and the presence of more severe radiographic signs of OA (i.e., effusion and osteophytosis) increases the risk and reduces the time until subsequent CrCL tear [48]. The contralateral stifles in OA dogs in this study did not show significant changes

in any of the SF biomarkers over the 12-week follow up period, and we were unable to predict the fate of the contralateral stifles based on the available follow-up data.

It has been postulated that presence of a complete CrCL tear results in more significant instability in the stifle and thereby may predispose the joint to a more significant degree of OA [49]. The main detectable change in this study was a statistically significant increase in the KC levels in the SF of dogs with a complete CrCL tear compared to stifles with partially torn CrCL. The observed increase in KC levels with complete CrCLR joints may be due to the higher degree of inflammatory response due to the extent of ligamentous damage. Concerns regarding exacerbation of OA secondary to meniscal damage have been expressed, and more severe OA is believed to result from the change in joint contact mechanics when the meniscus is lost [50]. In a canine meniscectomy model of OA, select SF cartilage biomarkers showed detectable alteration in these levels over time [19]. However, the current naturally occurring CrCLR dog model did not show any detectable differences in either serum or SF biomarkers between dogs with and without meniscal damage.

The histopathological grading system selected for this study showed a strong correlation between the sum of inflammatory infiltrate score and all three SF biomarkers, but not the serum biomarkers. However, MCP-1 was the only SF biomarker that had a strong correlation with all categories of this histological grading system. This histological grading system was proposed initially for use in rabbit synovial histological grading [28], and has not been validated for use in dogs. However, we selected this system to evaluate its ability to correlate with the biomarkers we evaluated because the subcategories proposed in other grading systems tended to be too broad to allow detailed categorization of the histological features [29, 40, 51]. Further, none of the other proposed grading systems have been validated for histological grading of canine synovitis. The proposed cut off values reported in this study, as well as the correlations amongst histological categories and the SF biomarkers, warrants further investigation and validation of the proposed use of this grading system in the histological grading of canine synovitis.

This study has several limitations due to the clinical nature of the OA model used. These limitations include the heterogenous population of dogs with variations in chronicity and severity of OA that may have resulted in lack of detection of significant differences between groups particularly in the serum biomarkers. It would be ideal if the chronicity of the CrCLR could further be stratified in future studies to evaluate potential differences in biomarkers over time. However, due to the variability of client-based reported chronicity of the CrCLR in the OA group and limitation of sample size this was not possible in the present study. All OA group dogs received a non-steroidal anti-inflamatory drug (NSAID) in the first 7–10 days after surgical intervention which may have impacted the results of follow up serum and SF biomarker measurements in this study. A previous study that has evaluated the effect of NSAID use on collagenase and general MMP activity in cartilage, and synovium did not show a significant difference compared to controls despite 8 weeks of use in a tCrCL model [52] In vitro studies have shown a dose-dependent suppressive effect by NSAIDs on MMPs in human synovial fibroblasts and bovine chondrocyte models [53, 54] as well as in clinical trials of human knee OA [55]. However, the impact of the suppressive effect of the short term meloxicam used in this study on the serum and SF biomarker measurements at T2 and T3 are unknown. Since all dogs in the OA groups were treated with the same protocol, the observed trends are expected to be similar within the OA group's temporal changes. The follow-up time in the study was relatively short (12 weeks) compared to the chronic, and insidious nature of OA progression, and was dictated by the limitations of utilizing client-owned animals for the purpose of the study. The synovial fluid samples were frozen after collection and were not spun until the time of analysis that may have affected relese of cytokines from the cells that may have affected the measured biomarkers. However, this methodology was based on the previous

study that had evaluated these biomarkers in a similar fashion and considering all samples in the current study were subjected to the same treatment, the overall effect of the freeze-thaw cycle on the level of cell lysis is expected to be consistent across all samples. The evaluated SF and serum biomarkers were compared between the control and OA associated with CrCLR, therefore the discriminatory ability of these markers in distinguishing between other forms of arthritis (e.g., septic, immune-mediated) cannot be determined based on the current study.

## Conclusions

Serum MMP-2 and MMP-3 concentrations did not discriminate between dogs with stifle OA secondary to naturally occurring CrCLR and controls, and cannot be recommended as clinically reliable serum biomarkers of stifle OA in this naturally occurring form. Of the investigated SF biomarkers, MCP-1 was the only biomarker with acceptable performance in discriminating between the OA group and controls. The significant decreases in IL-8, KC, and MCP-1 in the index stifles from the preoperative period to the 12-week recheck time point indicate an effect of surgical stabilization of a CrCL deficit knee, possibly in moderating the inflammatory response. Differences in the contralateral knee of dogs with unilateral CrCLR were detectable with elevations in KC levels in the joint when compared to controls and differences in IL-8 and MCP-1 levels compared to index stifle samples. These SF biomarker differences confirm the presence of a low-level existence of inflammation in the stable contralateral stifles of dogs with unilateral CrCLR. However, none of these SF biomarkers are suitable as predictors of future CrCLR in these initially stable stifles. MCP-1 was the only biomarker that consistently correlated with all categories of histopathologic grading, suggesting utility as a potential biomarker of synovial inflammation of canine stifle OA associated with CrCLR.

## Supporting information

**S1 Fig. Distribution of MMP-2 and MMP-3 serum concentrations between groups.** Distribution of MMP-2 (A) and MMP-3 (B) serum concentration for control and osteoarthritis (OA) groups (pg/ml). The horizontal line inside each box is the median and the upper and lower edges of box present the inter-quartile range (IQR). The whiskers are either $1.5 \times$ IQR or the range, whichever is smaller. Dots outside the fences are outliers.
(TIF)

**S2 Fig. Distribution of MMP-2 and MMP-3 serum concentrations over time in osteoarthritis (OA) group.** Distribution of serum concentrations of MMP-2 (A) and MMP-3 (B) over three time points for the OA group dogs. T1: initial visit, T2: 4-week recheck, T3: 12 week recheck. The horizontal line inside each box is the median and the upper and lower edges of box present the inter-quartile range (IQR). The whiskers are either $1.5 \times$ IQR or the range, whichever is smaller. Dots outside the fences are outliers. ** Statistically significant different concentrations ($P < 0.01$).
(TIF)

**S1 Data.**
(XLSX)

## Acknowledgments

The authors would like to thank Dr. Trina Bailey for contributing OA group samples, Dr. Aaron Stoker and Dr. Jimmy Cook for facilitating the multiplex bead assay analysis of the study samples at the University of Missouri's Comparative Orthopedic Laboratory.

## Author Contributions

**Conceptualization:** Sarah Malek, Romain Béraud, Christopher B. Riley.

**Data curation:** Sarah Malek, Mark C. Rochat, Romain Béraud.

**Formal analysis:** Sarah Malek, Hsin-Yi Weng, Shannon A. Martinson.

**Funding acquisition:** Sarah Malek, Romain Béraud, Christopher B. Riley.

**Investigation:** Sarah Malek, Shannon A. Martinson, Christopher B. Riley.

**Methodology:** Sarah Malek, Shannon A. Martinson, Romain Béraud, Christopher B. Riley.

**Project administration:** Sarah Malek, Romain Béraud, Christopher B. Riley.

**Resources:** Sarah Malek, Romain Béraud, Christopher B. Riley.

**Supervision:** Christopher B. Riley.

**Validation:** Sarah Malek.

**Writing – original draft:** Sarah Malek, Hsin-Yi Weng, Shannon A. Martinson, Mark C. Rochat, Christopher B. Riley.

**Writing – review & editing:** Sarah Malek, Hsin-Yi Weng, Shannon A. Martinson, Mark C. Rochat, Romain Béraud, Christopher B. Riley.

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
