## [Decision Letter · Decision Letter 0]

3 Jun 2020

PONE-D-20-13103

Evaluation of validity of serum MMP-2 and MMP-3, synovial fluid IL-8, MCP-1, and KC concentrations as biomarkers of stifle osteoarthritis associated with naturally occurring cranial cruciate ligament rupture in dogs

PLOS ONE

Dear Dr. Malek,

Thank you for submitting your manuscript to PLOS ONE. After careful consideration, we feel that it has merit but does not fully meet PLOS ONE’s publication criteria as it currently stands. Therefore, we invite you to submit a revised version of the manuscript that addresses the points raised during the review process.

We look forward to receiving your revised manuscript.

Kind regards,

Chi Zhang

Academic Editor

PLOS ONE

Journal Requirements:

2. In your Methods section, please provide additional details regarding the control dogs used in your study and ensure you have described the source and consent from the owners of the animals.

For more information regarding PLOS' policy on materials sharing and reporting, see https://journals.plos.org/plosone/s/materials-and-software-sharing#loc-sharing-materials.

"The funding for this study was secured through Canadian Institutes of Health Research Grant-Regional partnership fund - Innovation PEI (No: 97027) (CBR) (https://cihr-irsc.gc.ca/e/193.html), Companion Animal Trust Fund – University of Prince Edward Island (SM, RB) (https://www.upei.ca/avc/companion-animals/companion-animal-trust-fund), Cohn Family Chair for Small Animals- Oklahoma State University (SM) (https://news.okstate.edu/magazines/state-magazine/articles/2018/spring/cohn-family-chair-for-small-animals.html): direct and indirect costs and Boehringer-Ingelheim Ltd. financial incentive for client-owned dog recruitment by providing the non-steroidal anti-inflammatory pain medication meloxicam (Metacam®) (SM). The funders had no role in study design, data collection and analysis, decision to publish, or preparation of the manuscript."

We note that you received funding from a commercial source: Boehringer-Ingelheim Ltd.

4. We noted in your submission details that a portion of your manuscript may have been presented or published elsewhere.

"The results of this study regarding these biomarkers being evaluated as diagnostic and monitoring candidates for OA has been accepted as a poster at the 2020 OARSI’s  (osteoarthritis research society international) annual symposium) and will be published in abstract form in the osteoarthritis and cartilage journal in the near future. The findings of correlation of the histopathological data and the biomarkers have been accepted as a podium presentation at the 2020 annual ECVS (European College of Veterinary Surgeons) meeting and will be published as an abstract in the Veterinary Surgery Journal in the near future. Both these meetings have been cancelled due to the COVID-19 pandemic; therefore, the dat will only be published in abstract form in the aforementioned journals. However, this manuscript encompasses additional and expanded information from what were submitted to the ECVS and OARSI meeting."

Please clarify whether this conference proceeding / publication was peer-reviewed and formally published. If this work was previously peer-reviewed and published, in the cover letter please provide the reason that this work does not constitute dual publication and should be included in the current manuscript.

Reviewers' comments:

Reviewer's Responses to Questions

**Comments to the Author**

1. Is the manuscript technically sound, and do the data support the conclusions?

Reviewer #1: Yes

Reviewer #2: No

2. Has the statistical analysis been performed appropriately and rigorously? 

Reviewer #1: Yes

Reviewer #2: No

3. Have the authors made all data underlying the findings in their manuscript fully available?

Reviewer #1: Yes

Reviewer #2: Yes

4. Is the manuscript presented in an intelligible fashion and written in standard English?

Reviewer #1: Yes

Reviewer #2: Yes

5. Review Comments to the Author

Reviewer #1: The manuscript is very well written and for most part the data is also presented very well. It contains a substantial volume of work and some of the data presented are potentially important and may be useful in diagnosis and monitoring osteoarthritis in dogs. The background to the study, hypothesis and the study objectives are clearly stated however there are some minor issues that the authors need to address as detailed below.

Specific comments

1. There is a tendency to explain and discuss everything including non-significant findings! This makes it difficult to read and appreciate the main findings of the study. The authors could have written a more focused manuscript by concentrating on the important and significant data from the study. The authors should consider removing Figures 1 and 2 as these data are not significant and are given in Table 2. The description of the non-significant MMP-2 and MMP-3 results should be condensed as much as possible.

2. Figure 5 is an excellent summary of the main results and allows one to see at a glance the important and potentially useful findings of the entire study. Please provide a full description of this forest plot in the figure legend for readers who may not be familiar with this type of data presentation.

3. For Figures 3 and 4, an adequate description of the box plot is required. E.g. 95% CI, mean or median etc.

4. I enjoyed reading the discussion which is well written and balanced citing many appropriate references. However, it would be useful to add a short paragraph at the end of the discussion on the limitation and strength of the study.

Reviewer #2: The authors are quite clear in their abstract, introduction and methods that they set out to evaluate the validity of 5 markers (2 in serum [MMP2 and MMP3], 3 in synovial fluid (SF) [MCP-1, IL8, KC) as biomarkers, using a naturally occurring ACL deficiency in dogs who are having their ACL reconstructed as an OA group, representing a naturally occurring model of post-traumatic OA, and a control OA free group. Slightly ambitiously they appear to want to test the diagnostic ability, evaluate responsiveness to treatment and sensitivity to change in contralateral stifle joints and associations with other pathology. Despite this being an area of high clinical and scientific interest, there would appear to be some significant flaws which in my view affect the ability to draw clear conclusions. It’s also not really clear what clinical questions any of these 3 scenarios tested would usefully address in a situation where an animal was already known to be ACL deficient and was having reconstructive surgery.

MAJOR

Overall approach: Evaluation of biomarkers and assessment of validity are very specific FDA terms (l= https://www.ncbi.nlm.nih.gov/pmc/articles/PMC4430113/) with connotations in terms of approach. For example, for a diagnostic biomarker, this would need to be tested across several different disease groups to assess ‘specificity’ not just against a control group. This paper does not unfortunately get anywhere near achieving these aims. For example in terms of being a diagnostic marker for OA, all are likely to be upregulated in inflammatory arthritis or septic arthritis too, so it is not clear how they could be discriminatory if these groups were not examined. Similarly seeing the levels of these markers in OA which was not ACL deficient, to understand how specific they were to this post-traumatic OA phenotype, if that was the hypothesis here, would have been helpful. The hypothesis also included assessing predictive biomarkers but I see no real assessment of the role of these markers in prognosis.

Choice of makers: These 5 markers were picked from a paper by Garner et al, but it is not clear how relevant they are to these particular questions, or why these particular markers were selected when this original paper used just 10 animals. This perhaps just needed a little more justification.

Animals: I am unclear as to whether this is an acute injury situation (some have recently ruptured ACL) or a chronic ACL injury model or an established OA model. With the time from injury ranging from 1-732 days, this timing presumably could massively affect the injury vs OA response and hence the biomarker response, but it’s not clear if this was accounted for. Presumably the degree of OA must have varied hugely in these animals, and I wondered if we would be shown the biomarkers in comparison to this time from injury. There is just a comment that there were no significant differences. It is made clear this is not an animal model: as it is naturally occurring with a wide variety of breeds in both the OA group and control group with some neutered whilst others not brings yet more heterogeneity (detectable or not). It’s not clear why the control animals were being euthanased, and the role of use of meloxicam post operatively probably confounds looking at response to biomarkers (Bohringer-Ingelheim funded post op meloxicam I note in the financial disclosure)– was this a substudy in a clinical trial? This should have been made completely clear in methods.

Study design: The animals are sampled at 3 times – T1 initial, T2 4 weeks post op (ACL reconstruction), and T3, 12 weeks post op. The SF markers were found to drop over these times. However, the comparisons do not feel pre-defined, with multiple comparisons (albeit accounted for by Bonferroni) including comparison of contralateral and control joints. It is not clear how this could be used at an individual level to evaluate diagnosis or responsiveness to treatment, and in fact no clinical outcomes such as pain are measured in the study which seems an omission if one wants to look at responsiveness to treatment at an individual level. This is a shame because having longitudinal and contralateral SF sampling is a very significant resource. The study design, sampling and nomenclature is somewhat complex and could really have benefitted from a summary flow diagram.

Biomarker analysis: The synovial fluid was not spun at collection. This meant that any cells present (which are there) will lyse on freezing, causing artefact by release of cytokines including MCP-1 which could interfere with results. The biomarkers are analysed on a Luminex platform. Despite synovial fluid being well known to be a challenging fluid, no data is given as to the ‘validity’ of these assays on this matrix, or any performance characteristics of the assays, e.g. intra assay cv, interassay c.v. The synovial fluid was enzyme digested but this author would still need to be convinced that there was not lots of background in an assay like this.

Statistical analysis. This seems at odds to the initial questions. AUC is used, even to test serum makers which are statistically no different from controls. Later, contralateral limbs are included if they have OA and contralateral control limbs which causes issues with biological replicates in samples, which I am not convinced are fully accounted. Later in the histological analyses, Spearman’s correlation is used to look at an integer histological score and its relationship to the biomarkers which seems an odd approach. At one point, mean biomarker levels between the 2 limbs are calculated. This all seems fairly arbitrary and not clearly planned out. There is no discussion of whether the biomarkers are normally distributed and whether regression was the best approach. Some analyses are adjusted for covariates, but it is unclear what these are, whether they were predefined. They seem to include age and weight. Did they include the time from injury/instability (which would seem relevant) or breed of dog? Meniscal tear is mentioned for the first time at 311, with no detail on how this was recorded or accounted for. Imposing a cut off on MCP-1 seems like an afterthought and it’s not clear how much of this approach was pre-defined at the outset. The power calculation is not clear in terms of which question it was based on answering or whether this took in to account the plan to adjust for covariates (which I doubt given the very low estimated numbers needed).

Results. MMP3 is not significantly different. 5% reduction in levels is not an accurate way of describing this.

Discussion. There seems to be a lack of knowledge/citation of the human biomarker efforts in OA (Kraus et al, ARD 2017) and in PTOA relevant to this study (e.g. Struglics et al, A&R 2015). A knowledge of this literature might have informed some of the questions and approaches I suspect.

6. PLOS authors have the option to publish the peer review history of their article (what does this mean?). If published, this will include your full peer review and any attached files.

Reviewer #1: Yes: Mohammed Sharif

Reviewer #2: No

---

## [Author Response · Author response to Decision Letter 0]

17 Jul 2020

PONE-D-20-13103

Evaluation of serum MMP-2 and MMP-3, synovial fluid IL-8, MCP-1, and KC concentrations as potential biomarkers of stifle osteoarthritis associated with naturally occurring cranial cruciate ligament rupture in dogs

PLOS ONE

Dear Dr. Zhang and our reviewers, 

We would like to thank you all for your constructive feedback and recommendations. We have strived to answer all the questions and address all concerns raised and in some areas adjusted the manuscript content accordingly. The first section addresses comments from Dr. Zhang followed by our responses to each of our reviewers. Additional comments for Dr. Zhang are included in the cover letter for this revised manuscript. We hope that this version meets your expectations and that you find our responses to your questions and comments appropriate. 

Response to Academic Editor

 Authors’ response: Please let us know if we have missed any formatting requirements and we can readily address them. We apologize if we have missed more. The website indicates using Vancouver for output style for references, however, the endnote reference manage already has the formatting for Plos which actually matches your published work style. So, we used the plos output style from Endnote but we are happy to change that. 

Authors’ action: We found some errors in Table legends that were corrected and we spelled out the name USA in affiliations.

2. In your Methods section, please provide additional details regarding the control dogs used in your study and ensure you have described the source and consent from the owners of the animals.

For more information regarding PLOS' policy on materials sharing and reporting, see https://journals.plos.org/plosone/s/materials-and-software-sharing#loc-sharing-materials.

Authors’ response: Thank you for your comment. Obtaining approval from the control dogs’ population was not clear in the manuscript. These dogs were being euthanized at a shelter (as they are at many) for non-health related issues (e.g., behavioral issues, not being adoptable) and the shelter management was making the decision for euthanasia for the dogs and the euthanasia was conducted by the shelter staff. Dogs that were headed for euthanasia were screened for eligibility for inclusion in this study and permission by the shelter management was granted for all dogs that were included in this study. There were no financial or other incentive for the shelter management to allow the investigators to utilize these dogs. 

Authors’ action: We included that information in the statement in line 90 to confirm that consent was obtained for both population and with the owners’ understanding of all research related interventions the dogs were subjected to. We also added more detail in our ethics statement for the control dogs that would have been redundant for the body of the manuscript. 

"The funding for this study was secured through Canadian Institutes of Health Research Grant-Regional partnership fund - Innovation PEI (No: 97027) (CBR) (https://cihr-irsc.gc.ca/e/193.html), Companion Animal Trust Fund – University of Prince Edward Island (SM, RB) (https://www.upei.ca/avc/companion-animals/companion-animal-trust-fund), Cohn Family Chair for Small Animals- Oklahoma State University (SM) (https://news.okstate.edu/magazines/state-magazine/articles/2018/spring/cohn-family-chair-for-small-animals.html): direct and indirect costs and Boehringer-Ingelheim Ltd. financial incentive for client-owned dog recruitment by providing the non-steroidal anti-inflammatory pain medication meloxicam (Metacam®) (SM). The funders had no role in study design, data collection and analysis, decision to publish, or preparation of the manuscript."

We note that you received funding from a commercial source: Boehringer-Ingelheim Ltd.

 Authors’ response: The authors did not have any associations with the Boehringer-Ingelheim Ltd. regarding employment, consultancy, patents, products in development, marketed products, etc. As a non-steroidal anti-inflammatory pain medication, this drug is routinely prescribed to dogs after surgical intervention. The gift of the drugs was a donation of good will by the company in support of research at the authors’ facility (Atlantic Veterinary College). 

Authors’ action: We have addressed this as requested in our amended competing interests’ statement included as requested in the cover letter.

 Authors’ action: Done.

 4. We noted in your submission details that a portion of your manuscript may have been presented or published elsewhere.

"The results of this study regarding these biomarkers being evaluated as diagnostic and monitoring candidates for OA has been accepted as a poster at the 2020 OARSI’s (osteoarthritis research society international) annual symposium) and will be published in abstract form in the osteoarthritis and cartilage journal in the near future. The findings of correlation of the histopathological data and the biomarkers have been accepted as a podium presentation at the 2020 annual ECVS (European College of Veterinary Surgeons) meeting and will be published as an abstract in the Veterinary Surgery Journal in the near future. Both these meetings have been cancelled due to the COVID-19 pandemic; therefore, the data will only be published in abstract form in the aforementioned journals. However, this manuscript encompasses additional and expanded information from what were submitted to the ECVS and OARSI meeting."

Please clarify whether this conference proceeding / publication was peer-reviewed and formally published. If this work was previously peer-reviewed and published, in the cover letter please provide the reason that this work does not constitute dual publication and should be included in the current manuscript.

Authors’ response: The only published part of this work is the biomarker portion of the data that is produced in abstract form only in Osteoarthritis and Cartilage Journal (DOI: https://doi.org/10.1016/j.joca.2020.02.500). As previously mentioned neither the OARSI or ECVS meetings happened due to the COVID-19 pandemic, therefore no part of this work has been presented at any other venue. We also, retracted our second abstract prior to publication in Veterinary Surgery Journal in abstract form (this was the submission to the ECVS meeting). The abstract submitted to ECVS that had all the histopathological data and its correlations with the biomarker data, therefore, we do not believe this constitute dual publication for the current manuscript. 

Authors’ action: We have reflected this clarification in our cover letter as well. 

Response to reviewers’ comments:

Reviewer #1: The manuscript is very well written and for most part the data is also presented very well. It contains a substantial volume of work and some of the data presented are potentially important and may be useful in diagnosis and monitoring osteoarthritis in dogs. The background to the study, hypothesis and the study objectives are clearly stated however, there are some minor issues that the authors need to address as detailed below.

Specific comments

1. There is a tendency to explain and discuss everything including non-significant findings! This makes it difficult to read and appreciate the main findings of the study. The authors could have written a more focused manuscript by concentrating on the important and significant data from the study. The authors should consider removing Figures 1 and 2 as these data are not significant and are given in Table 2. The description of the non-significant MMP-2 and MMP-3 results should be condensed as much as possible.

Authors’ response: Thank you for your comments and your support of this manuscript. We agree that this work has a significant volume of information and can be cumbersome to read through, but we also felt that not reporting all findings on the key biomarkers that were under investigation will leave unanswered questions in the readers’ minds. The reason behind presenting Figure’s 1 and 2 (now renamed as S1 Fig and S2 Fig) was to give a visual aid for the readers regarding distribution of the biomarker concentrations between OA and control groups. The Figure 2 (now S2 Fig) does have a single significant point (T1 vs T2 for MMP2). The Table 2 is presented separately to demonstrate the details of the discriminative abilities rather than the comparisons of group means. We consider comparison of means (reported as difference in means, Lines 313-318) separately from the diagnostic discriminative abilities (reported based on AUC) (Lines 319-322). There are cases where group differences based on comparison of means may not be statistically significant but may provide valuable discriminative abilities. We also reported the magnitude of effect (i.e., difference in means and AUC to quantify clinical importance rather than simply reporting the P values. 

Authors’ action: We moved Figures 1 and 2 to supplementary section (now S1 Fig and S2 Fig).

2. Figure 5 is an excellent summary of the main results and allows one to see at a glance the important and potentially useful findings of the entire study. Please provide a full description of this forest plot in the figure legend for readers who may not be familiar with this type of data presentation.

Authors’ response: Thank you for your comments. The Figure 5 is now Figure 4 in the manuscript (Lines 412-420).

Authors’ action: Done. 

3. For Figures 3 and 4, an adequate description of the box plot is required. E.g. 95% CI, mean or median etc.

Authors’ response: Thank you for your comments.

Authors’ action: Done. The figures 3 and 4 are now figures 2 and 3 respectively. 

4. I enjoyed reading the discussion which is well written and balanced citing many appropriate references. However, it would be useful to add a short paragraph at the end of the discussion on the limitation and strength of the study.

Authors’ response: We completely agree with your feedback. We wanted this paragraph to be short, however, our second reviewer also provided us with very good points that needed to be clarified in this limitations section. Therefore, it is not as short as we would have hoped. 

Authors’ action: Limitation paragraph added in discussion (Lines 593-623) 

Reviewer #2: The authors are quite clear in their abstract, introduction and methods that they set out to evaluate the validity of 5 markers (2 in serum [MMP2 and MMP3], 3 in synovial fluid (SF) [MCP-1, IL8, KC) as biomarkers, using a naturally occurring ACL deficiency in dogs who are having their ACL reconstructed as an OA group, representing a naturally occurring model of post-traumatic OA, and a control OA free group. Slightly ambitiously they appear to want to test the diagnostic ability, evaluate responsiveness to treatment and sensitivity to change in contralateral stifle joints and associations with other pathology. Despite this being an area of high clinical and scientific interest, there would appear to be some significant flaws which in my view affect the ability to draw clear conclusions. It’s also not really clear what clinical questions any of these 3 scenarios tested would usefully address in a situation where an animal was already known to be ACL deficient and was having reconstructive surgery.

Authors’ response: Thank you for your detailed evaluation of this work. We would like to clarify that in the cohort of dogs with (CrCLR=ACL) deficiency in this study, the etiology of the rupture is not trauma as is seen in the human counterpart (PTOA), but a degenerative process. In dogs this results in loss of ACL’s structural integrity that is preceded by synovitis in the joint, resulting in partial or complete tear of the ligament during daily activities that do not necessarily fall under “traumatic” category. The etiology of this degenerative process remains elusive but has not been able to be attributed to any known immune mediated or infectious processes over the years. That is why dogs with history of traumatic ACL tear were not included in this study. This was not clear in our methodology and introduction for non-veterinarian readers who would not be as familiar with the process. The pathophysiology of this degenerative ACL disease in dogs affects many dogs bilaterally with around 50% of dogs experiencing a subsequent tear in the contralateral limb within a year of being diagnosed with a unilateral ACL tear. Therefore, our evaluation of the contralateral knee at different timelines was in pursuit of seeing whether we could detect any correlations between the fate of this initially stable knee and the biomarkers that would help us predict the ultimate outcome. We were also looking to detect early stages of the disease in the contralateral knee that may provide us with an opportunity for future treatments and preventative interventions. Due to the degenerative nature of the disease, attempts at reconstructing the native ACL in dogs with this particular pathology have not shown consistent good outcomes with the reconstructed ligament losing structural integrity due to the synovitis within the joint. Therefore in dogs, the majority of treatment options are aimed at mechanical stabilization of the joint by creating periarticular fibrosis using extracapsular femorotibial sutures or osteotomy techniques (e.g., TPLO) to adjust the tibial plateau slope (which is much steeper compared to humans in normal dogs) to provide dynamic stability of the joint in absence of the native ACL. Therefore regarding the contralateral stifles that were stable at the time of diagnosing CrCL in one knee, we were interested in mainly diagnostic ability of the synovial fluid biomarkers in detecting early OA changes compared to control group and sensitivity in changes over time but not responsiveness to treatment (since no treatment for the contralateral stifle was done). So in summary our hope for assessing the contralateral knees were: 

1- To see if these SF biomarkers show sensitivity in detecting these clinically stable knees that maybe already abnormal at a molecular level. This was with the idea that we may be able to use these markers for future studies in response to treatment or preventative interventions as a screening test to stop this degenerative process from resulting in CrCLR in these contralateral knees. 

2- We also wanted to see if we could predict based on these markers as to which contralateral knee would end up with CrCLR down the road after the initial diagnosis of unilateral CrCLR. 

Authors’ action: We added more detail in the introduction to clarify the significance of the naturally occurring CrCLR in dogs to point out the bilateral nature of the disease with subsequent tear rate being quite high in the initially stable contralateral knee (Lines 29-36). We expanded our 3rd objective to clarify our goals with the contralateral knee (Lines 81-82). We also added our exclusion of traumatic ACL tear dogs in methodology (Lines 110-111). In lines 135-137 in methods, we also emphasized that damaged portions were removed and no attempt at reconstructing the ligaments or menisci were made. 

MAJOR

Overall approach: Evaluation of biomarkers and assessment of validity are very specific FDA terms (l= https://www.ncbi.nlm.nih.gov/pmc/articles/PMC4430113/) with connotations in terms of approach. For example, for a diagnostic biomarker, this would need to be tested across several different disease groups to assess ‘specificity’ not just against a control group. This paper does not unfortunately get anywhere near achieving these aims. For example in terms of being a diagnostic marker for OA, all are likely to be upregulated in inflammatory arthritis or septic arthritis too, so it is not clear how they could be discriminatory if these groups were not examined. Similarly seeing the levels of these markers in OA which was not ACL deficient, to understand how specific they were to this post-traumatic OA phenotype, if that was the hypothesis here, would have been helpful. The hypothesis also included assessing predictive biomarkers but I see no real assessment of the role of these markers in prognosis.

Authors’ response: The clarification regarding this model not being post-traumatic but degenerative ACL tear has been addressed in our response to your preceding comment. We aimed to evaluate the sensitivity of these biomarkers in detecting the differences between the two groups. You are absolutely right that we did not look at specificity of these biomarkers for OA and drew correlations based on the fact that these dogs were systemically cleared of any other arthritic or systemic inflammatory disease prior to inclusion in the study. 

The evaluation of predictive biomarker role was investigated with regards to the fate of the contralateral stifles that were stable at the time of enrolment in the study was described in the methodology section (Lines 267-273). However, we did not find any correlations in the biomarkers (Lines 422-430). We did not believe that performing the same evaluation for the serum biomarkers would be useful considering we already had a unilateral ACL tear in one knee in those dogs and that serum biomarkers would be heavily affected by contributions from all joints but we did it anyways since we had the data and both are reported in the same paragraph in the results. 

Authors’ action: We removed the term “validity” from the manuscript’s title as well as the abstract and introduction and replaced them with evaluation and evaluate respectively. We also added a comment in our limitations to outline the fact that further specificity of these biomarkers for this model of OA needs to be evaluated against other causes of arthritis (Lines 639-642). We also added serum and SF in line 429 of results to show lack predictive value for these biomarkers for the contralateral CrCLR. 

Choice of makers: These 5 markers were picked from a paper by Garner et al, but it is not clear how relevant they are to these particular questions, or why these particular markers were selected when this original paper used just 10 animals. This perhaps just needed a little more justification.

Authors’ response: The main reason for this inclusion was that the Garner paper evaluated 18 biomarkers in serum synovial fluid and urine simultaneously and the majority of these markers had been previously evaluated in other studies with conflicting results and this study was able to show these 5 markers (2 in serum and 3 in synovial fluid) to be good candidates in both tCrCL model as well as the degenerative ACL model. The latter was the model we used in this study and the goal was to see if their results could be extrapolated to a larger and different population of dogs with the same pathology. The justification for inclusion of these markers are in the introduction (Lines 60-69).

Authors’ action: None.

Animals: I am unclear as to whether this is an acute injury situation (some have recently ruptured ACL) or a chronic ACL injury model or an established OA model. With the time from injury ranging from 1-732 days, this timing presumably could massively affect the injury vs OA response and hence the biomarker response, but it’s not clear if this was accounted for. Presumably the degree of OA must have varied hugely in these animals, and I wondered if we would be shown the biomarkers in comparison to this time from injury. There is just a comment that there were no significant differences. It is made clear this is not an animal model: as it is naturally occurring with a wide variety of breeds in both the OA group and control group with some neutered whilst others not brings yet more heterogeneity (detectable or not). 

Authors’ response: On the veterinary side, we are so used to the fact that there is an obvious distinction between traumatic ACL tear and the degenerative (naturally-occurring) ACL tear that we had not clarified that in our methodology section adequately. Due to the degenerative nature of this disease and the variability of the speed with which the clinical instability of the stifle occurs, we were not able to control for chronicity of the rupture in our recruitment strategy. However, we did evaluate the impact of chronicity on our results by arbitrarily selecting less than 30 days being acute and more than 30 days being chronic for when the owners first reported clinical signs of pain and lameness. However, you are correct that this only roughly singles out the ACL tear timeline and not necessarily development of OA. We therefore agree that it creates more confusion plus we didn’t find any significant difference by this categorization based on symptoms. The breeds of dogs included in this study are all previously reported to be prone to developing ACL tears and with lack balance in the number of breeds and presence of mixed breed dogs, evaluating effect of breed was not possible. We also discussed inclusion of sex initially as a covariate but due to the skewedness of the data and the fact that in dogs effect of sex hormones on OA has had reportedly mixed findings, we elected to not use it in our analysis. This latter had been stated in lines 290-291. 

Authors’ action: We have added history of traumatic CrCL tear (Line 110-111) as part of the exclusion criteria to clarify this. We also added a comment in our limitations to point out that it would be ideal if we could have a clear idea when these tears initiated to be able to see if stratification of these cases based on chronicity would show any differences that we were unable to detect in this study (Lines 594-600). We also removed evaluation of chronicity from our objectives and our methods and reported results. 

It’s not clear why the control animals were being euthanized, and the role of use of meloxicam post operatively probably confounds looking at response to biomarkers (Bohringer-Ingelheim funded post op meloxicam I note in the financial disclosure)– was this a substudy in a clinical trial? This should have been made completely clear in methods.

Authors’ response: The dogs were being culled in a shelter setting based on the shelter management’s decision for various causes (not being adoptable, mostly behavioral issues). Therefore, the decision to euthanize had nothing to do with the current study but we were given permission to sample these dogs that were singled out for euthanasia. We have clarified this in our ethics statement, but did not find it necessary to expand on it beyond the comment of “euthanized for reasons unrelated to this project” in Line 119 in the body of the manuscript. 

We completely agree with your statement that meloxicam introduced a bias in the study. However, from an animal ethical standpoint (including institutional Animal Ethics Committee approval) we could not withhold the standard of care during the postoperative period for these patients which requires use of a form of NSAID. We do believe that this was mainly affecting the T2 and T3 timelines and not the initial visit measurements. However, we agree that this has to be clearly pointed out. We also added a section in our financial disclosure. The donation of the meloxicam was a goodwill act on the company’s behalf that not only provided the owners with an incentive but also allowed us to have a homogenous pain management regiment postoperatively. From the company’s marketing standpoint, this was helpful for owner of the dogs to be able to use this drug over other NSAIDs in the veterinary market. However, the Boehringer company did not have any involvement in the study design, execution, data analysis, or data reporting. 

Authors’ action: We have clarified the recruitment and fate of control dogs in our ethics statement. We have added comments regarding potential impact of use of meloxicam on the results of this study (Lines 600-611)

Study design: The animals are sampled at 3 times – T1 initial, T2 4 weeks post op (ACL reconstruction), and T3, 12 weeks post op. The SF markers were found to drop over these times. However, the comparisons do not feel pre-defined, with multiple comparisons (albeit accounted for by Bonferroni) including comparison of contralateral and control joints. It is not clear how this could be used at an individual level to evaluate diagnosis or responsiveness to treatment, and in fact no clinical outcomes such as pain are measured in the study which seems an omission if one wants to look at responsiveness to treatment at an individual level. This is a shame because having longitudinal and contralateral SF sampling is a very significant resource. 

Authors’ response: Thank you for your comments. We were mainly interested in the ability of surgical stabilization of the knee (we did not reconstruct the ACL in any of these dogs as this is not a common or generally successful treatment option in dogs with degenerative ACL tear) in reducing these inflammatory biomarkers in the joint. We agree that it would have been great to correlate these findings with patient findings but there are no reliable direct and objective outcome measures of orthopedic pain in veterinary medicine, and the most commonly used measure of clinical pain are some of the client questionnaires that tend to have significant bias and placebo effect associated with their results. We had assessed overall pain in these patients using the CBPI (canine brief pain inventory) questionnaire that is the most commonly used measure of pain (care giver assessment) but had not included the results in the study due to the fact that studies have shown that the results of the CBPI do not correlate with functional improvements as measured by more reliable outcome measures such as force plate gait analysis. All of these patients improved in their ambulatory status following surgery and the instability in their operated stifles resolved and achieved mechanical stability in the operated knees. However, due to the subjectivity of the pain assessment in clinical patients, further correlation of the biomarker data with the CBPI questionnaire results were not pursued. The limitation of this study for outcome measure assessment was our inability to conduct any force plate gait analysis in an attempt to correlate biomarker changes over time with the force plate data. The additional complexity in this trial was also the bilateral nature of the disease in many of the included patients. Therefore, our focus was on pure measurements of the biomarkers and evaluating temporal changes in both CrCLR stifles and the contralateral stable stifles over time. 

Authors’ action: None. 

The study design, sampling and nomenclature is somewhat complex and could really have benefitted from a summary flow diagram.

Authors’ response: Thank you for your comments. We worked on making all references to the OA (index) stifle and contralateral (stable) stifles more clear. It was not possible to show the index versus contralateral on the flow chart since some of these dogs were bilaterally and some unilaterally affected with CrCLR

Authors’ action: The nomenclature has been defined in methodology section particularly regarding the CrCLR and contralateral samples in Lines 169-177. We had also clarified the T1-T3 time points in lines 146-150. We hope that the diagram in Figure 1 that was added helps further clarify this. 

Biomarker analysis: The synovial fluid was not spun at collection. This meant that any cells present (which are there) will lyse on freezing, causing artefact by release of cytokines including MCP-1 which could interfere with results. 

Authors’ response: We did not perform spinning of the synovial fluid samples for two reasons. Firstly, we used the remainder of the fluid for a different study that required the unchanged sample. Secondly, the Garner et al. study also did not perform spinning of the samples at the time of collection. However, all samples were treated the same, and therefore cells that could have released these biomarkers due to freezing, were potentially actively producing them for release from the cell to the synovial fluid. Therefore, it is likely that the contribution of the lysed cells to the concentration of the cytokine in the synovial fluid is likely proportional to the cells contribution of the cytokine concentration prior to freezing. 

Authors’ action: We have added a comment in the limitation section of our discussion to discuss this point (lines 615-621). 

The biomarkers are analysed on a Luminex platform. Despite synovial fluid being well known to be a challenging fluid, no data is given as to the ‘validity’ of these assays on this matrix, or any performance characteristics of the assays, e.g. intra assay cv, interassay c.v. The synovial fluid was enzyme digested but this author would still need to be convinced that there was not lots of background in an assay like this.

Authors’ response: The Luminex platform used in this study has been previously tested specifically on synovial fluid samples and have been shown to be reliable (https://scholar.google.com/scholar?hl=en&as_sdt=0%2C26&q=Luminex+Assay+on+synovial+fluid&btnG=). An example of this for intra-assay precision is in the following publication in Osteoarthritic and Cartilage journal: https://www.oarsijournal.com/action/showPdf?pii=S1063-4584%2813%2900209-4 . We did not have any concerns regarding interassy performance since our comparisons were within the same assay in this study. 

Authors’ action: None. 

Statistical analysis. This seems at odds to the initial questions. AUC is used, even to test serum makers which are statistically no different from controls. Later, contralateral limbs are included if they have OA and contralateral control limbs which causes issues with biological replicates in samples, which I am not convinced are fully accounted.

Authors’ response: We used comparison of groups’ means and AUC to address different research questions. Comparison of means was to answer whether central locations of two populations were different, whereas AUC was to quantify diagnostic discriminative performance of the biomarkers. Thus, we reported both sets of results regardless of statistical significance. In addition, we also reported the magnitude of effect (i.e., difference in means and AUC) to quantify clinical importance rather than simply reporting the p values, which are affected by both effect size and sample size. 

Authors’ action: clarified the results in comparison of means more in the results (Lines 314-319).

Later in the histological analyses, Spearman’s correlation is used to look at an integer histological score and its relationship to the biomarkers which seems an odd approach.

Authors’ response: The Spearman’s correlation is used as histopathological grading is measured at an ordinal scale. 

Authors’ action: None.

 At one point, mean biomarker levels between the 2 limbs are calculated. This all seems fairly arbitrary and not clearly planned out. There is no discussion of whether the biomarkers are normally distributed and whether regression was the best approach.

Authors’ response: The serum comparisons do not involve any OA and contralateral comparisons as the sample is from the shared systemic circulation. However, for the synovial fluid samples that came directly from each stifle, the comparisons were relevant. We compared the two synovial samples of each control dog (right and left) and since we did not find any clinically important and statistically significant difference, we used the mean of each dog’s right and left biomarker measurements to perform the ROC analysis. This was in an attempt to avoid duplicate samples from each control dog but rather having a single value for each SF biomarker for each control dog without completely omitting one sample (lines 396-400). 

The S1 Figure & S2 Figure in supplementary and Fig 2 and 3 in the manuscript show the data distributions for all biomarkers based on boxplots. As indicated in these figures, some distributions are right-skewed and thus log transformation was applied. The regression methods used to analyze the data are robust to normality assumption. In addition, these variables were measured at an interval scale and the sample size were relatively sufficient, therefore, central theorem could be applied. In summary, the selected regression models are optimal methods to account for covariates (i.e., in the analyses of serum biomarkers) and/or dependency of observations (i.e., in the analyses of repeated measures and synovia fluid biomarkers) in this study. 

 Some analyses are adjusted for covariates, but it is unclear what these are, whether they were predefined. They seem to include age and weight. Did they include the time from injury/instability (which would seem relevant) or breed of dog? Meniscal tear is mentioned for the first time at 311, with no detail on how this was recorded or accounted for. 

Authors’ response: We defined using adjustments for significant covariates in line 252 but did not name age or weight specifically, since we looked at all possible factors that could be significant and of those that were or where possible to use, age and weight were the influential ones. We reported age and weight in results in lines 286-288 as having impact and being used to adjust for during comparisons. Evaluation of meniscal tear in operated stifles had been defined in lines 132-135. The result section reports the number of torn menisci and degree of tear in the ACL in lines 304-311. The chronicity of the ACL tear was evaluated (time from injury) and in lines 301-306) we report that due to lack of difference between the acute and chronic dogs based on our categorization and as we responded to your similar comment earlier, we removed this from our analysis because it is only a relatively decent measure for chronicity of ACL tear rather than OA. We did not include breed due to the accuracy of classification and limited number in some of the breeds and large number of mixed breed dogs. 

Author’s action: added via arthrotomy and arthroscopy in methodology in line 132 and added at the time of surgery in line 135 for meniscal evaluation. 

Imposing a cut off on MCP-1 seems like an afterthought and it’s not clear how much of this approach was pre-defined at the outset.

Authors’ response: The cut-off value was not an afterthought but mainly based on the fact that MCP-1 was the only biomarker that yielded an AUC of ≥0.90, therefore a cut-off value was statistically determined based on the Youden’s index. This analytical approach had been pre-defined in lines 262-265. 

 The power calculation is not clear in terms of which question it was based on answering or whether this took in to account the plan to adjust for covariates (which I doubt given the very low estimated numbers needed).

Authors’ response: The power analysis question was based on MMP2 and MMP3 differences between OA and control group and we did not take into account adjustments for covariates in our power analysis. Historically, age and weight effect on OA changes in dogs have shown contradictory results in previous published research. Therefore, unlike in people, they are not well-established standard covariates that are accounted for in canine research. Although we agree that they should be in the future considering some of our results as we used them in our analysis but had not accounted for them in our initial power analysis. 

Authors’ action: We added MMP-2 and MMP-3 to clarify the power calculation (line 94). 

Results. MMP3 is not significantly different. 5% reduction in levels is not an accurate way of describing this.

Authors’ response: By 5% we are describing the magnitude of the difference in group means. We stated these differences to quantify clinical importance, as p value does not directly measure clinical importance. A 5% reduction in the mean MMP-3 concentration in the OA group compared to control groups indicates a small difference, whereas a 2.4 fold higher mean of MMP-2 concentration in the OA group indicates a much larger difference, despite that both were not statistically significant. 

Authors’ action: We reworded our presentation of these results to help avoid any misinterpretation of the information (Lines 315-319).

Discussion. There seems to be a lack of knowledge/citation of the human biomarker efforts in OA (Kraus et al, ARD 2017) and in PTOA relevant to this study (e.g. Struglics et al, A&R 2015). A knowledge of this literature might have informed some of the questions and approaches I suspect.

Authors’ response: Thank you for your comments. We are familiar with the extensive work on the human side with biomarkers of OA, particularly based on the PTOA, however, the focus of this study was to evaluate the biomarkers in relation to other canine biomarker work and to draw references to the human literature only where we lacked canine data or found contradictory results that were more in line with some of the human literature. The references you have cited in your comments are by far superior to the canine study here due to the larger sample size, inclusion of some well-established human biomarkers, and the extent of follow up. However, we believe adding more citation of the human studies in an attempt to fully review and compare the body of work on these biomarkers in the current manuscript would have resulted in an even lengthier manuscript. We are aware of the work performed by these teams including some of the work of the Struglics group with even longer follow up times using the PTOA model (e.g., Newman et al. 10.1016/j.joca.2016.09.008). We have therefore, included this limitation of lack of long-term follow up in our limitations (Lines 612-615). The radiographic grading of OA in humans have various validated diagnostic biomarkers (e.g., joint space narrowing) as well as self-reported pain assessments that are valuable outcome measures. However, on the clinical canine case side there are no validated radiographic measurements of OA that correlate with severity of OA, which leaves room for more research on the veterinary side. We attempted to use one of the more commonly used previously reported radiographic grading system (Innes et al. 2004; 10.1111/j.1740-8261.2004.04024.x ). However, the grades were too broad preventing us from being able to perform meaningful statistical analysis. Therefore, we did not include that in our study but rather used the radiographs to confirm presence of OA, and lack of other radiographic abnormalities in the joints. 

Authors’ action: Included lack of long-term follow up in our limitations (Lines 612-615). 

6. PLOS authors have the option to publish the peer review history of their article (what does this mean?). If published, this will include your full peer review and any attached files.

Do you want your identity to be public for this peer review? For information about this choice, including consent withdrawal, please see our Privacy Policy.

Reviewer #1: Yes: Mohammed Sharif

Reviewer #2: No

---

## [Decision Letter · Decision Letter 1]

29 Sep 2020

PONE-D-20-13103R1

Evaluation of serum MMP-2 and MMP-3, synovial fluid IL-8, MCP-1, and KC concentrations as biomarkers of stifle osteoarthritis associated with naturally occurring cranial cruciate ligament rupture in dogs

PLOS ONE

Dear Dr. Malek,

Thank you for submitting your manuscript to PLOS ONE. After careful consideration, we feel that it has merit but does not fully meet PLOS ONE’s publication criteria as it currently stands. Therefore, we invite you to submit a revised version of the manuscript that addresses the points raised during the review process.

We look forward to receiving your revised manuscript.

Kind regards,

Chi Zhang

Academic Editor

PLOS ONE

Reviewers' comments:

Reviewer's Responses to Questions

**Comments to the Author**

1. If the authors have adequately addressed your comments raised in a previous round of review and you feel that this manuscript is now acceptable for publication, you may indicate that here to bypass the “Comments to the Author” section, enter your conflict of interest statement in the “Confidential to Editor” section, and submit your "Accept" recommendation.

Reviewer #3: All comments have been addressed

Reviewer #4: (No Response)

2. Is the manuscript technically sound, and do the data support the conclusions?

Reviewer #3: Yes

Reviewer #4: Yes

3. Has the statistical analysis been performed appropriately and rigorously? 

Reviewer #3: Yes

Reviewer #4: Yes

4. Have the authors made all data underlying the findings in their manuscript fully available?

Reviewer #3: No

Reviewer #4: Yes

5. Is the manuscript presented in an intelligible fashion and written in standard English?

Reviewer #3: Yes

Reviewer #4: Yes

6. Review Comments to the Author

Reviewer #3: The authors have thoroughly considered the many comments from the reviewers, and in my opinion made appropriate adjustments and corrections. The nature of the study makes it exploratory - all overstatements have been reformulated. Also excess figure/tables have been moved to supplementary, giving the manuscript a better flow

Reviewer #4: I believe that the authors have addressed the limitations previously highlighted by reviewers. I have noted that the authors took into account the recommendations to be more cautious in describing the markers they are highlighting as having a discriminatory value rather than using diagnostic, particularly in the discussion. I therefore would like to recommend that they use this terminology consistently when describing their objectives and throughout the paper.

7. PLOS authors have the option to publish the peer review history of their article (what does this mean?). If published, this will include your full peer review and any attached files.

Reviewer #3: No

Reviewer #4: No

---

## [Author Response · Author response to Decision Letter 1]

27 Oct 2020

Dear Dr. Zhang and our reviewers, 

We would like to thank you all for your constructive feedback and recommendations. We hope that this version meets your expectations and that you find our responses to your comments appropriate. 

Response to Academic Editor

We did not see any specific comments from you regarding the manuscript’s content itself. However, we are looking into seeing if the protocols.io applies to us and will be getting our protocol added there. We have included the tracked and untracked version of the manuscript along with all the supplementary files and figures. Our financial disclosure and ethics statement has not changed. It appears that our reviewers did not have significant concerns during this review; therefore, our response to their comments is limited to the only point raised in section “6. Review comments to the author”. We also followed the recommended PACE software to check the figure requirements, and Figure 1 was the only file that was adjusted by the system. We uploaded the updated version in the new submission. We also added the data spreadsheet in a compressed format as a supplementary file to fulfill the data availability requirement of the journal. 

Response to reviewers’ comments:

We have revised the check box to allow data availability in response to reviewer 3’s answer. 

Reviewer #3: The authors have thoroughly considered the many comments from the reviewers, and in my opinion made appropriate adjustments and corrections. The nature of the study makes it exploratory - all overstatements have been reformulated. Also excess figure/tables have been moved to supplementary, giving the manuscript a better flow. 

Authors’ response: We very much appreciate your feedback and are happy that our efforts have met your expectations. 

Authors’ action: None. 

Reviewer #4: I believe that the authors have addressed the limitations previously highlighted by reviewers. I have noted that the authors took into account the recommendations to be more cautious in describing the markers they are highlighting as having a discriminatory value rather than using diagnostic, particularly in the discussion. I therefore would like to recommend that they use this terminology consistently when describing their objectives and throughout the paper.

Authors’ response: Thank you very much for your positive feedback. We agree that the diagnostic phrase can be misleading, as we did not evaluate these biomarkers in differentiating this form of OA from other arthritic conditions. We appreciate you noticing this issue throughout the paper. 

Authors’ action: We have removed the term “diagnostic” when referring to the current biomarkers’ performance and ability and have simply used the term discriminative performance/ability throughout the paper. We have highlighted areas where the term “diagnostic” was removed. We chose the term discriminative over discriminatory to be more particular in the use of the word as an adjective.

---

## [Editor Report · Decision Letter 2]

6 Nov 2020

Evaluation of serum MMP-2 and MMP-3, synovial fluid IL-8, MCP-1, and KC concentrations as biomarkers of stifle osteoarthritis associated with naturally occurring cranial cruciate ligament rupture in dogs

PONE-D-20-13103R2

Dear Dr. Malek,

We’re pleased to inform you that your manuscript has been judged scientifically suitable for publication and will be formally accepted for publication once it meets all outstanding technical requirements.

Kind regards,

Chi Zhang

Academic Editor

PLOS ONE

---

## [Editor Report · Acceptance letter]

10 Nov 2020

PONE-D-20-13103R2 

Evaluation of serum MMP-2 and MMP-3, synovial fluid IL-8, MCP-1, and KC concentrations as biomarkers of stifle osteoarthritis associated with naturally occurring cranial cruciate ligament rupture in dogs 

Dear Dr. Malek:

I'm pleased to inform you that your manuscript has been deemed suitable for publication in PLOS ONE. Congratulations! Your manuscript is now with our production department. 

Kind regards, 

on behalf of

Dr. Chi Zhang 

Academic Editor

PLOS ONE